# MOTION GENERALIST: MULTIMODAL MOTION GENERATION

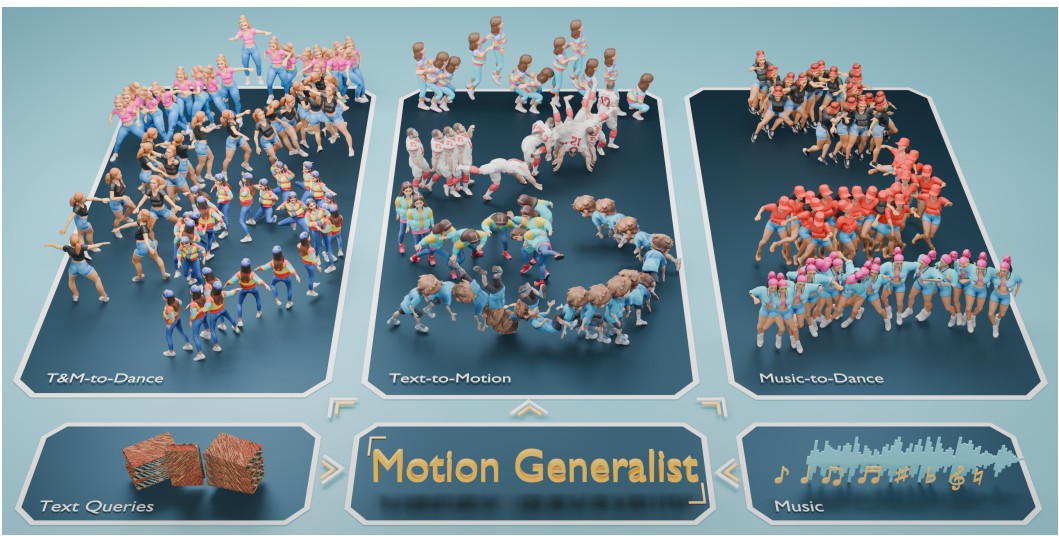

Figure 1: *Motion Generalist* is an versatile method for generating high-quality, controllable human motion under multimodal conditions, including text queries, background music and a mix of both.

## ABSTRACT

Conditional motion generation has been extensively studied in computer vision, yet two critical challenges remain. First, while masked autoregressive methods have recently outperformed diffusion-based approaches, existing masking models lack a mechanism to prioritize dynamic frames and body parts based on given conditions. Second, existing methods for different conditioning modalities often fail to integrate multiple modalities effectively, limiting control and coherence in generated motion. To address these challenges, we propose **Motion Generalist**, a multimodal motion generation framework that introduces an Attention-based Mask Modeling approach, enabling fine-grained spatial and temporal control over key frames and actions. Our model adaptively encodes multimodal conditions, including text and music, improving controllability. Additionally, we introduce **Text-Music-Dance (TMD)**, a new motion dataset consisting of **2,153** pairs of text, music, and dance, making it **twice** the size of AIST++, thereby filling a critical gap in the community. Extensive experiments demonstrate that Motion Generalist surpasses state-of-the-art methods across multiple benchmarks, achieving a **15%** improvement in FID on HumanML3D and showing consistent performance gains on AIST++ and TMD.

## 1 INTRODUCTION

Human motion generation (Zhu et al., 2023) has been widely explored in recent years due to its broad applications in film production, video gaming, augmented and virtual reality (AR/VR), and embodied AI for human-robot interaction. Recent advancements in conditional motion generation, including *text-to-motion* (Pinyoanuntapong et al., 2024b; Yuan et al., 2024; Hosseyni et al., 2024) and *music-to-dance* (Siyao et al., 2022; Li et al., 2024b) models have shown promising potential

in 3D motion generation. These developments mark significant progress in generating motion sequences directly from textual descriptions and background music. However, despite extensive research in motion generation, the field still faces two significant challenges.

(1) Recently, masked autoregressive methods (Pinyoanuntapong et al., 2024a; Guo et al., 2024) have shown a promising trend, outperforming diffusion-based methods (Tevet et al., 2022; Chen et al., 2023; Zhang et al., 2023b). However, existing masking models have been underexplored in generating motion that prioritizes dynamic frames and body parts in motion sequences based on given conditions.

(2) Although specialized and multitask methods (Gong et al., 2023; Zhou & Wang, 2023; Zhang et al., 2024e; Bian et al., 2024) exist for different conditioning modalities, they often overlook the importance of integrating multiple modalities to achieve more controllable generation, as shown in Table 1. For example, enhancing music-to-dance generation with precise text descriptions can improve control and coherence, whereas relying on only a single modality as a condition leads to underperformance.

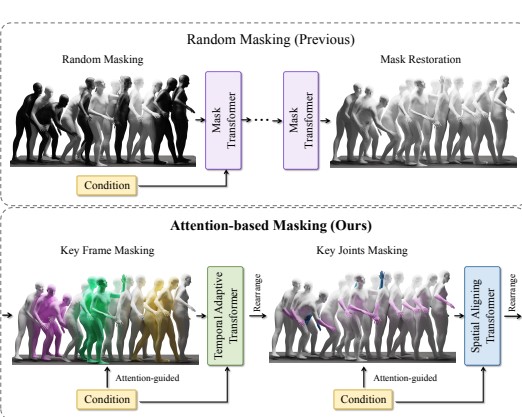

Figure 2: **Masking strategy comparison.** The previous random masking (Guo et al., 2024) (top) vs. our attention-based masking (bottom).

Our motivation is to address these challenges by presenting an innovative method that tackles them in a nutshell. To overcome the first challenge, we designed a conditional masking approach within an autoregressive generation paradigm across both spatial and temporal dimensions, enabling the model to focus on key frames and actions corresponding to the given condition, as shown in Figure 2. The conditional masking strategy also dynamically adjusts based on the modality of the condition, whether it is text or music. To tackle the second challenge, we design our architecture to handle multimodal conditions adaptively and simultaneously. From a temporal perspective, our model aligns different input modalities to control motion generation in a time-sensitive manner. Meanwhile, from a spatial perspective, it maps action queries to specific body-part movements and aligns music genres with corresponding dance styles. Moreover, since multi-conditioning in motion generation is underexplored, there is no motion dataset with paired music and text available in the current community. Hence, we have curated a new motion dataset with paired music and text as a benchmark to help advance the community's exploration of multimodal conditioning in motion generation.

In general, our contributions can be summarized as follows:

- We present *Motion Generalist*, an versatile framework that can seamlessly and adaptively encode multimodal conditions for more controllable motion generation. This fills the gap of multimodal conditioning in previous motion generation research and represents a significant improvement.
- For generating more controllable motion, we design an *Attention-based Mask Modeling* approach across both temporal and spatial dimensions, focusing on key frames and key actions corresponding to the condition. We further customize the mask transformer to adaptively handle different modalities of conditions, enhancing motion generation by integrating multimodal conditioning.
- For exploring multi-conditioning motion generation, we introduce the new *Text-Music-Dance* (**TMD**) dataset, which includes **2,153** paired samples of text, music, and dance, making it **twice** as large as AIST++ (Li et al., 2021). We also conducted extensive experiments on standard benchmarks across multiple motion generation tasks. Our method achieved a **15%** improvement in FID on HumanML3D (Guo et al., 2022a) and consistent improvement on AIST++ (Li et al., 2021) and TMD datasets.

## 2 RELATED WORKS

**Text-to-Motion Generation.** Recent advancements in human motion generation have skillfully combined diffusion and autoregressive models, achieving more realistic, versatile, and scalable motion synthesis. Foundational work like MDM (Tevet et al., 2022) introduced a transformer-based

Table 1: **Methods comparison.** Either single-task or multi-task models can handle only one condition at a time, overlooking the importance of integrating multiple modalities for more controllable generation. Our Motion Generalist introduces an innovative approach that encodes different modalities simultaneously and adaptively for more controllable generation.

| Models | Text-to-Motion | Music-to-Dance | Text and Music to Dance |
|---|:---:|:---:|:---:|
| TM2D (Gong et al., 2023) | ✓ | ✓ | ✗ |
| UDE (Zhou & Wang, 2023) | ✓ | ✓ | ✗ |
| UDE-2 (Zhou et al., 2023b) | ✓ | ✓ | ✗ |
| MoFusion (Dabral et al., 2023) | ✓ | ✓ | ✗ |
| MCM (Ling et al., 2024) | ✓ | ✓ | ✓ |
| LMM (Zhang et al., 2024e) | ✓ | ✓ | ✗ |
| MotionCraft (Bian et al., 2024) | ✓ | ✓ | ✗ |
| MagicPose4D (Zhang et al., 2024c) | ✗ | ✗ | ✗ |
| STAR (Chai et al., 2024) | ✓ | ✗ | ✗ |
| TC4D (Bahmani et al., 2025) | ✓ | ✗ | ✗ |
| Motion Avatar (Zhang et al., 2024i) | ✓ | ✗ | ✗ |
| **Motion Generalist (Ours)** | ✓ | ✓ | ✓ |

diffusion approach for lifelike, text-driven motion generation. Expanding on this, MotionDiffuse (Zhang et al., 2024d) added refined control and diversity mechanisms, while MLD (Chen et al., 2023) boosted efficiency by operating within a latent space, reducing computational demands without sacrificing quality. Motion Mamba (Zhang et al., 2025) addressed the challenge of generating longer sequences, and ReMoDiffuse (Zhang et al., 2023b) further enriched motion variability by incorporating retrieval-augmented diffusion. Meanwhile, autoregressive models like MoMask (Guo et al., 2024) enhanced temporal coherence through generative masked modeling, selectively revealing segments of the motion sequence. BAMM (Pinyoanuntapong et al., 2024a) introduced a bidirectional model to capture detailed motion with forward and backward dependencies. Infini-Motion (Zhang et al., 2024h) optimized transformer memory to support extended sequences, and KMM (Zhang et al., 2024g) prioritized essential frames to balance continuity and computational efficiency. MoGenTS (Yuan et al., 2024) added spatial-temporal joint modeling for further structural consistency in generated motions.

**Music-to-Dance Generation.** Recent work in music-driven dance generation has leveraged autoregressive and diffusion-based models to achieve more synchronized, diverse, and controllable dance motions. TSMT (Li et al., 2020) pioneered using transformer architectures to model complex dance motions. Subsequently, early methods like DanceNet (Zhuang et al., 2022) and Dance Revolution (Huang et al., 2021) customize autoregressive and sequence-to-sequence models to establish foundational mappings between music and movement. FACT (Li et al., 2021) and Bailando (Siyao et al., 2022) build upon this by incorporating 3D motion data and actor-critic memory models to capture richer choreography, and Bailando++ (Siyao et al., 2023) enhances this framework further for refined generation quality. EDGE (Tseng et al., 2023) introduces user-editable dance generation for greater customization. In recent work, Lodge (Li et al., 2024b) and Lodge++ (Li et al., 2024a) apply coarse-to-fine diffusion methods to extend sequence length and create vivid choreography patterns, while Beat-It (Huang et al., 2024b) achieves beat-synchronized dance generation under multiple musical conditions. Lastly, the BADM (Zhang et al., 2024a) merges autoregressive and diffusion models, producing coherent, music-aligned dance sequences. Together, these works illustrate the field's progression toward high-fidelity dance generation tightly integrated with musical features.

**Multi-Task Motion Generation.** Human motion generation has evolved through multi-modal approaches, enabling contextually adaptive synthesis across diverse inputs like music, text, and visual cues. TM2D (Gong et al., 2023) introduced a bimodal framework integrating music and text for 3D dance generation, using VQ-VAE to encode motion in a shared latent space for flexible control. MotionCraft (Bian et al., 2024) builds on this by offering whole-body motion generation with adaptable multi-modal controls, from high-level semantics to specific joint details. MCM (Ling et al., 2024) further employs a transformer-based model to process varied inputs—text, audio, and video—generating motion that reflects both style and context. LMM (Zhang et al., 2024e) extends these capabilities by integrating large-scale pre-trained models for complex human motion across modalities. Meanwhile, UDE (Zhou & Wang, 2023) provides a cohesive framework for motion synthesis, and UDE-2 (Zhou et al., 2023b) expands this to synchronize multi-part, multi-modal movements. MoFusion (Dabral et al., 2023) complements these advances with a diffusion-based denoising framework focused on robustness and quality in diverse motion styles. These works collectively guide the field toward adaptive, multi-modal, and context-aware human motion generation.

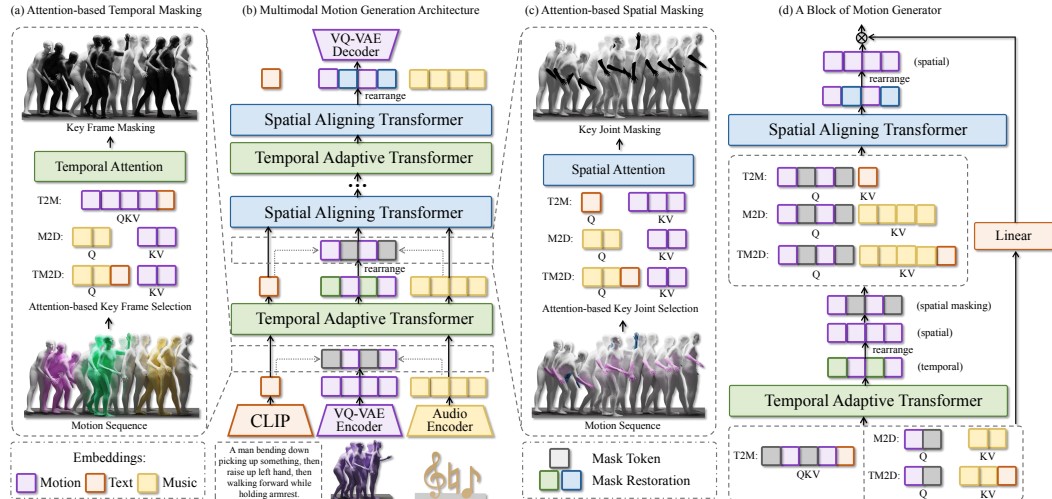

Figure 3: **Motion Generalist architecture.** The multimodal architecture consists of several key components: (a) temporal and (c) spatial attention-based masking, (b) motion generator, and (d) a single block of motion generator. These components enable the model to learn key motions corresponding to the given conditions, and facilitate alignment between multi-modal conditions and motion features.

# 3 METHODOLOGY

## 3.1 OVERVIEW

Motion Generalist presents an innovative versatile approach that generates controllable human motion by focusing on the dynamic and significant parts of human motion sequences and adaptively aligning with different condition modalities. As shown in Figure 3, Motion Generalist can take different modalities either separately or simultaneously, enabling multimodal conditioning to enhance controllable motion generation instead of relying on a single condition. The conditions are first encoded by text and audio encoders, then used to guide both masking and motion generation. We propose an attention-based masking approach that iden-

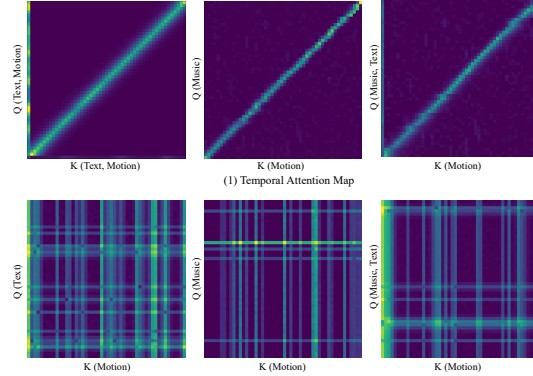

Figure 4: **Attention map.** The attention map provides a direct visualization of our attention-based masking approach, which selectively masks regions in the motion sequence with high attention scores.

tifies the most significant parts of the motion corresponding to the conditions across both spatial and temporal dimensions by selecting high-attention scores as masking guidance. The guided mask tokens, along with condition embeddings, are then fed into a masked transformer for guided mask restoration. We customize masked transformers into a Temporal Adaptive Transformer and a Spatial Aligning Transformer to adaptively align overall control and specific actions to the motion sequence.

## 3.2 ARCHITECTURE

**Attention-based Masking.** The core of attention-based masking involves guiding the condition modality to select key frames in the temporal dimension and key actions in the spatial dimension, allowing for the masking of these motions. As shown in the attention map in Figure 4, both temporal and spatial attention rely on either self-attention or cross-attention (Vaswani, 2017), depending on the condition modality. The condition serves as query $Q$, and motion serves as key $K$ and value $V$, where the condition can be text, audio, or a combination of both. This process highlights specific regions in the attention map, indicating the key motions. We designed attention-based masking on both temporal and spatial dimensions to ensure the model focuses on learning key frames and joints in the motion sequence that correspond to the conditions, as shown in Figure 3 (a) and (c).

This enables the model to learn more robust motion representations compared to traditional random masking (Guo et al., 2024; Yuan et al., 2024).

As shown in Appendix B Algorithm 1, given a motion sequence $\mathbf{M}$, a condition $\mathbf{C}$, and a masking ratio $\alpha$, the model masks the top $\alpha\%$ of attention scores, which represent the most important motions that are also most relevant to the corresponding condition.

**Temporal Adaptive Transformer.** The Temporal Adaptive Transformer (TAT) aligns the temporal tokens of the motion sequence with the temporal condition by dynamically adjusting its attention calculation according to the modality of the condition. This enables the TAT to align key frames of motion with keywords in text and beats in music.

As shown in Appendix B Algorithm 2 and Figure 3 (d), after attention-based masking, the key frames of the motion sequence are masked. The TAT then learns the motion representation by restoring the masked frames with guidance from the condition. If the condition consists only of text, it con-

Table 2: **Quantitative comparison on AIST++ (Li et al., 2021).** The best and runner-up values are **bold** and underlined. Multimodal motion generation methods are highlighted in blue.

| Method | Motion Quality | | Motion Diversity | | |
|---|---|---|---|---|---|
| | $\mathrm{FID}_k \downarrow$ | $\mathrm{FID}_g \downarrow$ | $\mathrm{Div}_k \uparrow$ | $\mathrm{Div}_g \uparrow$ | BAS $\uparrow$ |
| Ground Truth | 17.10 | 10.60 | 8.19 | 7.45 | 0.2374 |
| TSMT (Li et al., 2020) | 86.43 | 43.46 | 6.85 | 3.32 | 0.1607 |
| Dance Revolution (Huang et al., 2021) | 73.42 | 25.92 | 3.52 | 4.87 | 0.1950 |
| DanceNet (Zhuang et al., 2022) | 69.18 | 25.49 | 2.86 | 2.85 | 0.1430 |
| MoFusion (Dabral et al., 2023) | 50.31 | - | 9.09 | - | 0.2530 |
| EDGE (Tseng et al., 2023) | 42.16 | 22.12 | 3.96 | 4.61 | 0.2334 |
| Lodge (Li et al., 2024b) | 37.09 | 18.79 | 5.58 | 4.85 | 0.2423 |
| FACT (Li et al., 2021) | 35.35 | 22.11 | 5.94 | 6.18 | 0.2209 |
| Bailando (Siyao et al., 2022) | 28.16 | 9.62 | 7.83 | 6.34 | 0.2332 |
| TM2D (Gong et al., 2023) | 23.94 | 9.53 | 7.69 | 4.53 | 0.2127 |
| BADM (Zhang et al., 2024a) | - | - | 8.29 | 6.76 | 0.2366 |
| LMM (Zhang et al., 2024e) | 22.08 | 21.97 | 9.85 | 6.72 | 0.2249 |
| Bailando++ (Siyao et al., 2023) | 17.59 | 10.10 | 8.64 | 6.50 | 0.2720 |
| UDE (Zhou & Wang, 2023) | 17.25 | 8.69 | 7.78 | 5.81 | 0.2310 |
| MCM (Ling et al., 2024) | **15.57** | 25.85 | 6.50 | 5.74 | 0.2750 |
| **Motion Generalist (Ours)** | 17.22 | **8.56** | **9.91** | **6.79** | **0.2757** |

tains a single token in the temporal dimension from CLIP, making self-attention in the temporal dimension more suitable. Otherwise, the motion sequence serves as $Q$, and the condition serves as $KV$, performing cross-attention to align the temporal information of the motion with music or the combination of music and text. This enables the Temporal Adaptive Transformer to become more adaptable and robust for different modalities of input conditions.

**Spatial Aligning Transformer.** In the Spatial Aligning Transformer (SAT), both the condition and motion embeddings are rearranged to expose the spatial dimension. As shown in Appendix B Algorithm 3 and Figure 3 (d), during attention-based masking, the key action in each frame, which refers to the key motion of a specific body part in the spatial dimension, is masked. The SAT restores this feature with the guidance of the spatial condition. Aligning the spatial pose in each frame with the spatial condition is essential, especially in text-to-motion generation, where certain keywords describe specific body parts. In music-to-dance generation, the spectrum of each audio frame indicates the music genre (Tzanetakis & Cook, 2002; Lee et al., 2009), which is crucial for generating the appropriate type of dance.

## 4 EXPERIMENTS

### 4.1 DATASETS AND EVALUATION METRICS

**TMD Dataset.** Our Text-Music-Dance (TMD) dataset introduces a pioneering benchmark with 2,153 pairs of text, music, and motion. We extract dance motions and corresponding text annotations from Motion-X (Lin et al., 2024), including AIST++ (Li et al., 2021) and other datasets. For motion-text pairs without music, we generate corresponding music by implementing Stable Audio Open (Evans et al., 2024) with beat adjustment and evaluate the generated music through human expert assessments, ensuring inter-rater reliability.

**Public Benchmarks.** To ensure a fair comparison, we evaluate our method against both specialized and unified motion generation approaches on standard benchmarks including HumanML3D (Guo et al., 2022a) and KIT-ML (Plappert et al., 2016) for text-to-motion generation, and AIST++ (Li et al., 2021) for music-to-dance generation.

**Evaluation Metrics.** We adapt standard evaluation metrics to assess various aspects of our experiments. For text-to-motion generation, we implement FID and R precision to quantify the realism and robustness of generated motions, MultiModal Distance to measure motion-text alignment, and the

Table 3: **Quantitative comparison on HumanML3D (Guo et al., 2022a) and KIT-ML (Plappert et al., 2016).** The best and runner-up values are **bold** and underlined. The right arrow → indicates that closer values to ground truth are better. Multimodal motion generation methods are highlighted in blue.

| Datasets | Method | R Precision ↑ | | | FID↓ | MultiModal Dist↓ | Diversity→ | MultiModality↑ |
|---|---|---|---|---|---|---|---|---|
| | | Top 1 | Top 2 | Top 3 | | | | |
| | Ground Truth | $0.511^{\pm.003}$ | $0.703^{\pm.003}$ | $0.797^{\pm.002}$ | $0.002^{\pm.000}$ | $2.974^{\pm.008}$ | $9.503^{\pm.065}$ | - |
| | TM2D (Gong et al., 2023) | $0.319^{\pm.000}$ | - | - | $1.021^{\pm.000}$ | $4.098^{\pm.000}$ | $9.513^{\pm.000}$ | $4.139^{\pm.000}$ |
| | MotionCraft (Bian et al., 2024) | $0.501^{\pm.003}$ | $0.697^{\pm.003}$ | $0.796^{\pm.002}$ | $0.173^{\pm.002}$ | $3.025^{\pm.008}$ | $9.543^{\pm.098}$ | - |
| | ReMoDiffuse (Zhang et al., 2023b) | $0.510^{\pm.005}$ | $0.698^{\pm.006}$ | $0.795^{\pm.004}$ | $0.103^{\pm.004}$ | $2.974^{\pm.016}$ | $9.018^{\pm.075}$ | $1.795^{\pm.043}$ |
| Human | MMM (Pinyoanuntapong et al., 2024b) | $0.504^{\pm.003}$ | $0.696^{\pm.003}$ | $0.794^{\pm.002}$ | $0.080^{\pm.003}$ | $2.998^{\pm.007}$ | $9.411^{\pm.058}$ | $1.164^{\pm.041}$ |
| ML3D | DiverseMotion (Lou et al., 2023) | $0.515^{\pm.003}$ | $0.706^{\pm.002}$ | $0.802^{\pm.002}$ | $0.072^{\pm.004}$ | $2.941^{\pm.007}$ | $9.683^{\pm.102}$ | $1.869^{\pm.089}$ |
| (Guo et al., 2022a) | BAD (Hosseyni et al., 2024) | $0.517^{\pm.002}$ | $0.713^{\pm.003}$ | $0.808^{\pm.003}$ | $0.065^{\pm.003}$ | $2.901^{\pm.008}$ | $9.694^{\pm.068}$ | $1.194^{\pm.044}$ |
| | BAMM (Pinyoanuntapong et al., 2024a) | $0.525^{\pm.002}$ | $0.720^{\pm.003}$ | $0.814^{\pm.006}$ | $0.055^{\pm.002}$ | $2.919^{\pm.008}$ | $9.717^{\pm.089}$ | $1.687^{\pm.051}$ |
| | MCM (Ling et al., 2024) | $0.502^{\pm.002}$ | $0.692^{\pm.004}$ | $0.788^{\pm.006}$ | $0.053^{\pm.007}$ | $3.037^{\pm.003}$ | $9.585^{\pm.082}$ | $0.810^{\pm.023}$ |
| | MoMask (Guo et al., 2024) | $0.521^{\pm.002}$ | $0.713^{\pm.002}$ | $0.807^{\pm.002}$ | $0.045^{\pm.002}$ | $2.958^{\pm.008}$ | - | $1.241^{\pm.040}$ |
| | LMM (Zhang et al., 2024e) | $0.525^{\pm.002}$ | $\underline{0.719}^{\pm.002}$ | $0.811^{\pm.002}$ | $0.040^{\pm.002}$ | $2.943^{\pm.012}$ | $9.814^{\pm.076}$ | $2.683^{\pm.054}$ |
| | MoGenTS (Yuan et al., 2024) | $\underline{0.529}^{\pm.003}$ | $\underline{0.719}^{\pm.002}$ | $\underline{0.812}^{\pm.002}$ | $\underline{0.033}^{\pm.001}$ | $\underline{2.867}^{\pm.006}$ | $9.570^{\pm.077}$ | - |
| | **Motion Generalist (Ours)** | $\mathbf{0.546}^{\pm.003}$ | $\mathbf{0.735}^{\pm.002}$ | $\mathbf{0.829}^{\pm.002}$ | $\mathbf{0.028}^{\pm.005}$ | $\mathbf{2.859}^{\pm.010}$ | $\underline{9.521}^{\pm.083}$ | $\underline{2.705}^{\pm.068}$ |
| | Ground Truth | $0.424^{\pm.005}$ | $0.649^{\pm.006}$ | $0.779^{\pm.006}$ | $0.031^{\pm.004}$ | $2.788^{\pm.012}$ | $11.08^{\pm.097}$ | - |
| | ReMoDiffuse (Zhang et al., 2023b) | $0.427^{\pm.014}$ | $0.641^{\pm.004}$ | $0.765^{\pm.055}$ | $0.155^{\pm.006}$ | $2.814^{\pm.012}$ | $10.80^{\pm.105}$ | $1.239^{\pm.028}$ |
| | MMM (Pinyoanuntapong et al., 2024b) | $0.404^{\pm.005}$ | $0.621^{\pm.005}$ | $0.744^{\pm.004}$ | $0.316^{\pm.028}$ | $2.977^{\pm.019}$ | $10.91^{\pm.101}$ | $1.232^{\pm.039}$ |
| KIT- | DiverseMotion (Lou et al., 2023) | $0.416^{\pm.005}$ | $0.637^{\pm.008}$ | $0.760^{\pm.011}$ | $0.468^{\pm.098}$ | $2.892^{\pm.041}$ | $10.87^{\pm.101}$ | $\mathbf{2.062}^{\pm.079}$ |
| ML | BAD (Hosseyni et al., 2024) | $0.417^{\pm.006}$ | $0.631^{\pm.006}$ | $0.750^{\pm.006}$ | $0.221^{\pm.025}$ | $2.941^{\pm.012}$ | $11.00^{\pm.100}$ | $1.170^{\pm.047}$ |
| (Plappert et al., 2016) | BAMM (Pinyoanuntapong et al., 2024a) | $0.438^{\pm.009}$ | $0.661^{\pm.009}$ | $0.788^{\pm.005}$ | $0.183^{\pm.013}$ | $2.723^{\pm.026}$ | $\mathbf{11.01}^{\pm.094}$ | $1.609^{\pm.065}$ |
| | MoMask (Guo et al., 2024) | $0.433^{\pm.007}$ | $0.656^{\pm.005}$ | $0.781^{\pm.005}$ | $0.204^{\pm.011}$ | $2.779^{\pm.022}$ | - | $1.131^{\pm.043}$ |
| | LMM (Zhang et al., 2024e) | $0.430^{\pm.015}$ | $0.653^{\pm.017}$ | $0.779^{\pm.014}$ | $\underline{0.137}^{\pm.023}$ | $2.791^{\pm.018}$ | $11.24^{\pm.103}$ | $\underline{1.885}^{\pm.127}$ |
| | MoGenTS (Yuan et al., 2024) | $\underline{0.445}^{\pm.006}$ | $\underline{0.671}^{\pm.006}$ | $\underline{0.797}^{\pm.005}$ | $0.143^{\pm.004}$ | $\underline{2.711}^{\pm.024}$ | $10.92^{\pm.090}$ | - |
| | **Motion Generalist (Ours)** | $\mathbf{0.449}^{\pm.007}$ | $\mathbf{0.678}^{\pm.004}$ | $\mathbf{0.802}^{\pm.006}$ | $\mathbf{0.131}^{\pm.003}$ | $\mathbf{2.705}^{\pm.024}$ | $10.94^{\pm.098}$ | $1.374^{\pm.069}$ |

Table 4: **Quantitative comparison on TMD.** The best and runner-up values are **bold** and underlined.

| Method | Motion Quality | | Motion Diversity | | BAS ↑ | MMDist↓ | MModality↑ |
|---|---|---|---|---|---|---|---|
| | FID$_k$ ↓ | FID$_g$ ↓ | Div$_k$ ↑ | Div$_g$ ↑ | | | |
| Ground Truth | 20.72 | 11.37 | 7.42 | 6.94 | 0.2105 | 5.07 | - |
| TM2D (Gong et al., 2023) | 26.78 | $\underline{12.04}$ | 6.25 | 4.41 | 0.2001 | 6.13 | 2.232 |
| MotionCraft (Bian et al., 2024) | $\underline{24.21}$ | 26.39 | $\underline{7.02}$ | $\underline{5.79}$ | $\underline{0.2036}$ | $\underline{5.82}$ | **2.481** |
| **Motion Generalist** | **21.46** | **11.44** | **7.04** | **6.15** | **0.2094** | **5.34** | $\underline{2.424}$ |

diversity metric to calculate variance in motion features. Additionally, we apply the multi-modality (MModality) metric to evaluate diversity among motions sharing the same text description. For music-to-dance generation, we follow AIST++ (Li et al., 2021) to evaluate generated dances from three perspectives: quality, diversity, and music-motion alignment. For quality, we calculate FID between the generated dance and motion sequence features (kinetic, FID$_k$, and geometric, FID$_g$) using the toolbox in (Gopinath & Won, 2020). For diversity, we compute the average feature distance as in AIST++ (Li et al., 2021). For alignment, we calculate the Beat Align Score (BAS) as the average temporal distance between music beats and their closest dance beats.

## 4.2 MODEL AND IMPLEMENTATION DETAILS

Our model consists of 2 TAT and 2 SAT layers, with 12.65M parameters and 137.35 GFLOPs. The learning rate increases to $2 \times 10^{-4}$ after 2000 iterations using a linear warm-up schedule for all models. The mini-batch size is set to 512 for training the VQ-VAE tokenizer and 64 for training the masked transformers. All experiments were conducted on an Intel Xeon Platinum 8360Y CPU at 2.40GHz, equipped with a single NVIDIA A100 40GB GPU and 32GB of RAM.

## 4.3 COMPARATIVE STUDY

**Text-to-Motion.** We compared our method with other state-of-the-art approaches on both HumanML3D (Guo et al., 2022a) and KIT-ML (Plappert et al., 2016). The results in Table 3 demonstrate that our method consistently outperforms specialized text-to-motion models and surpasses recent multi-task methods.

**Music-to-Dance.** To highlight the music-to-dance capability of our method, we conducted evaluations on AIST++ (Li et al., 2021). The results in Table 2 indicate that our method surpasses previous state-of-the-art specialized and unified approaches, demonstrating superior motion quality, enhanced diversity, and better beat alignment in music-to-dance generation.

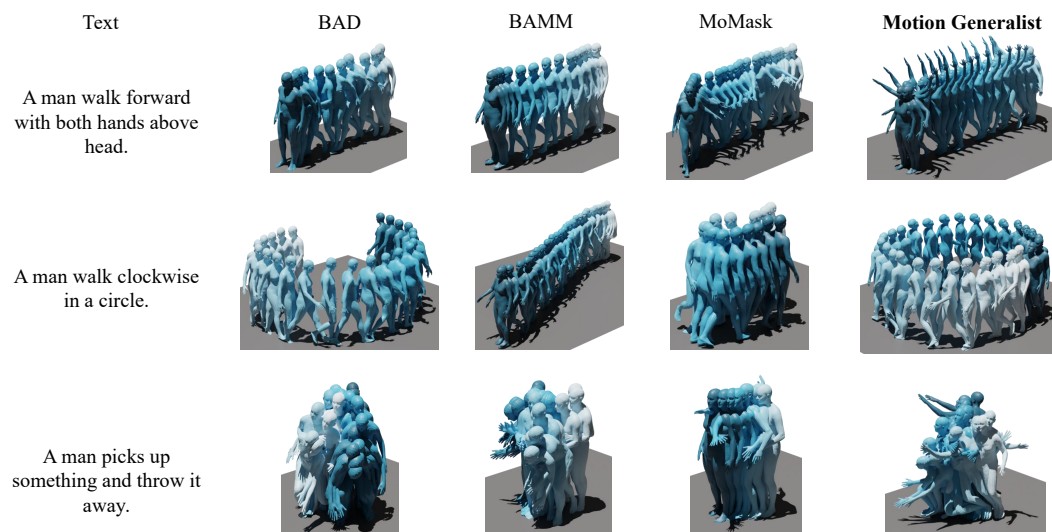

|  | Text | BAD | BAMM | MoMask | **Motion Generalist** |

Figure 5: **Qualitative evaluation on text-to-motion generation.** We qualitatively compared the visualizations generated by our method with those produced by BAD (Hosseyni et al., 2024), BAMM (Pinyoanuntapong et al., 2024a), and MoMask (Guo et al., 2024).

Table 5: **Ablation study of the masking strategy on HumanML3D (Guo et al., 2022a).** The best and runner-up values are **bold** and underlined. The right arrow → indicates that closer values to ground truth are better.

| Method | R Precision ↑ | | | FID↓ | MM Dist↓ | Diversity→ | MModality↑ |
|---|---|---|---|---|---|---|---|
|  | Top 1 | Top 2 | Top 3 |  |  |  |  |
| Ground Truth | $0.511^{\pm.003}$ | $0.703^{\pm.003}$ | $0.797^{\pm.002}$ | $0.002^{\pm.000}$ | $2.974^{\pm.008}$ | $9.503^{\pm.065}$ | - |
| Random Masking (Guo et al., 2024) | $0.522^{\pm.004}$ | $0.714^{\pm.003}$ | $0.818^{\pm.006}$ | $0.049^{\pm.023}$ | $2.945^{\pm.027}$ | $9.633^{\pm.218}$ | $2.538^{\pm.035}$ |
| KMeans (Lloyd, 1982) | $0.528^{\pm.003}$ | $0.709^{\pm.004}$ | $0.823^{\pm.006}$ | $0.042^{\pm.032}$ | $\underline{2.871}^{\pm.035}$ | $9.549^{\pm.173}$ | $2.548^{\pm.023}$ |
| GMM (Reynolds et al., 2009) | $0.531^{\pm.002}$ | $0.721^{\pm.004}$ | $\underline{0.826}^{\pm.008}$ | $0.039^{\pm.021}$ | $2.887^{\pm.024}$ | $9.602^{\pm.138}$ | $2.488^{\pm.031}$ |
| Confidence-based Masking (Pinyoanuntapong et al., 2024b) | $0.524^{\pm.007}$ | $0.731^{\pm.001}$ | $0.818^{\pm.004}$ | $0.047^{\pm.023}$ | $2.928^{\pm.009}$ | $9.530^{\pm.095}$ | $2.574^{\pm.039}$ |
| Density-based Masking (Zhang et al., 2024g) | $\underline{0.538}^{\pm.005}$ | $\underline{0.733}^{\pm.002}$ | $0.819^{\pm.006}$ | $\underline{0.031}^{\pm.035}$ | $2.913^{\pm.021}$ | $\mathbf{9.518}^{\pm.138}$ | $\underline{2.608}^{\pm.043}$ |
| **Attention-based Masking** | $\mathbf{0.546}^{\pm.003}$ | $\mathbf{0.735}^{\pm.002}$ | $\mathbf{0.829}^{\pm.002}$ | $\mathbf{0.028}^{\pm.005}$ | $\mathbf{2.859}^{\pm.010}$ | $\underline{9.521}^{\pm.083}$ | $\mathbf{2.705}^{\pm.068}$ |

**Text-and-Music-to-Dance.** For paired text-and-music-to-dance (TM2D) generation, we evaluated open-source multimodal motion generation methods by directly combining their condition embeddings and compared them with our approach on the TMD dataset. The results in Table 4 demonstrate superior performance by our method when handling different modalities simultaneously.

## 4.4 ABLATION STUDY

**Masking Strategy.** To evaluate the effectiveness of our attention-based masking approach across both temporal and spatial dimensions, we conducted comprehensive experiments on HumanML3D (Guo et al., 2022a), comparing it to other masking strategies such as random masking (Guo et al., 2024; Yuan et al., 2024), KMeans (Lloyd, 1982), GMM (Reynolds et al., 2009), confidence-based masking (Pinyoanuntapong et al., 2024b), and density-based masking (Zhang et al., 2024g). The results in Table 5 demonstrate that our attention-based masking outperforms these other strategies in human motion generation, yielding promising results for learning robust motion representations.

**Masking Ratio.** To demonstrate the robustness of our method across various hyperparameters and the impact of different masking ratios on overall performance, we conducted comprehensive ablation studies with different attention-based masking ratios on HumanML3D (Guo et al., 2022a), as shown in Table 6. We conducted an ablation study on the masking ratio for temporal and spatial attention-based masking separately. The results show that our method is relatively robust across different masking ratios, with 30% identified as the superior setting in our paper.

**Cross-modal TAT for Text-to-Motion.** To verify the necessity of self-attention in the Temporal Adaptive Transformer (TAT) for text-to-motion generation, we modified a cross-modal attention

Table 6: **Ablation study of masking ratio on HumanML3D (Guo et al., 2022a).** The best and runner-up values are **bold** and underlined. The right arrow → indicates that closer values to ground truth are better.

| Method | R Precision ↑ | | | FID↓ | MM Dist↓ | Diversity→ | MModality↑ |
|---|---|---|---|---|---|---|---|
| | Top 1 | Top 2 | Top 3 | | | | |
| Ground Truth | $0.511^{\pm.003}$ | $0.703^{\pm.003}$ | $0.797^{\pm.002}$ | $0.002^{\pm.000}$ | $2.974^{\pm.008}$ | $9.503^{\pm.065}$ | - |
| T:15% S:15% | $0.523^{\pm.005}$ | $0.716^{\pm.002}$ | $0.818^{\pm.005}$ | $0.047^{\pm.034}$ | $2.920^{\pm.026}$ | $9.625^{\pm.145}$ | $2.580^{\pm.064}$ |
| T:15% S:30% | $0.529^{\pm.002}$ | $0.718^{\pm.005}$ | $0.822^{\pm.003}$ | $0.044^{\pm.046}$ | $2.914^{\pm.023}$ | $9.573^{\pm.163}$ | $2.631^{\pm.024}$ |
| T:15% S:50% | $0.530^{\pm.002}$ | $0.715^{\pm.007}$ | $0.820^{\pm.007}$ | $0.045^{\pm.035}$ | $2.918^{\pm.019}$ | $9.632^{\pm.217}$ | $2.611^{\pm.026}$ |
| T:30% S:15% | $0.535^{\pm.007}$ | $\underline{0.728}^{\pm.001}$ | $\underline{0.823}^{\pm.004}$ | $0.036^{\pm.027}$ | $\underline{2.873}^{\pm.037}$ | $\mathbf{9.527}^{\pm.116}$ | $\underline{2.709}^{\pm.027}$ |
| **T:30% S:30%** | $\mathbf{0.546}^{\pm.003}$ | $\mathbf{0.735}^{\pm.002}$ | $\mathbf{0.829}^{\pm.002}$ | $\mathbf{0.028}^{\pm.005}$ | $\mathbf{2.859}^{\pm.010}$ | $\underline{9.521}^{\pm.083}$ | $2.705^{\pm.068}$ |
| T:30% S:50% | $\underline{0.541}^{\pm.004}$ | $0.726^{\pm.003}$ | $0.821^{\pm.005}$ | $\underline{0.033}^{\pm.035}$ | $2.926^{\pm.054}$ | $9.519^{\pm.196}$ | $\mathbf{2.710}^{\pm.037}$ |
| T:50% S:15% | $0.525^{\pm.005}$ | $0.720^{\pm.003}$ | $0.820^{\pm.009}$ | $0.043^{\pm.028}$ | $2.940^{\pm.044}$ | $9.620^{\pm.134}$ | $2.584^{\pm.063}$ |
| T:50% S:30% | $0.525^{\pm.007}$ | $0.723^{\pm.004}$ | $0.819^{\pm.007}$ | $0.040^{\pm.042}$ | $2.937^{\pm.063}$ | $9.617^{\pm.115}$ | $2.701^{\pm.031}$ |
| T:50% S:50% | $0.524^{\pm.006}$ | $0.712^{\pm.003}$ | $0.822^{\pm.006}$ | $0.048^{\pm.025}$ | $2.943^{\pm.037}$ | $9.623^{\pm.153}$ | $2.620^{\pm.025}$ |

Table 7: **Ablation study of the TAT on HumanML3D (Guo et al., 2022a).** The best values are **bold**. The right arrow → indicates that closer values to ground truth are better.

| Method | R Precision ↑ | | | FID↓ | MM Dist↓ | Diversity→ | MModality↑ |
|---|---|---|---|---|---|---|---|
| | Top 1 | Top 2 | Top 3 | | | | |
| Ground Truth | $0.511^{\pm.003}$ | $0.703^{\pm.003}$ | $0.797^{\pm.002}$ | $0.002^{\pm.000}$ | $2.974^{\pm.008}$ | $9.503^{\pm.065}$ | - |
| Cross-modal Attention | $0.347^{\pm.006}$ | $0.587^{\pm.007}$ | $0.726^{\pm.005}$ | $0.583^{\pm.024}$ | $3.356^{\pm.022}$ | $9.032^{\pm.153}$ | $2.153^{\pm.056}$ |
| **Motion Generalist** | $\mathbf{0.546}^{\pm.003}$ | $\mathbf{0.735}^{\pm.002}$ | $\mathbf{0.829}^{\pm.002}$ | $\mathbf{0.028}^{\pm.005}$ | $\mathbf{2.859}^{\pm.010}$ | $\mathbf{9.521}^{\pm.083}$ | $\mathbf{2.705}^{\pm.068}$ |

Table 8: **Single-modal vs. multimodal generation** on TMD dataset.

| | Motion Quality | | Motion Diversity | | | | |
|---|---|---|---|---|---|---|---|
| Method | $\text{FID}_k$ ↓ | $\text{FID}_g$ ↓ | $\text{Div}_k$ ↑ | $\text{Div}_g$ ↑ | BAS ↑ | MMDist↓ | MModality↑ |
| Ground Truth | 20.72 | 11.37 | 7.42 | 6.94 | 0.2105 | 5.07 | - |
| Motion Generalist w/o text | 25.07 | 14.23 | 6.95 | 6.01 | 0.2077 | 6.24 | 2.398 |
| **Motion Generalist** | **21.46** | **11.44** | **7.04** | **6.15** | **0.2094** | **5.34** | **2.424** |

layer in TAT to resemble the setup implemented for music-to-dance and text&music-to-dance generation. The results in Table 7 indicate that the cross-modal attention layer performs worse compared to the self-attention layer in TAT for text-to-motion on HumanML3D (Guo et al., 2022a). This underperformance can be attributed to the fact that the text embeddings from CLIP consist of only a single token along the temporal dimension. Consequently, the temporal dimension does not align with the motion embeddings, making it unsuitable for effective cross-modal fusion with motion in the temporal context.

**Effectiveness of Multimodal Conditioning.** To evaluate the effectiveness of multimodal conditioning, we examine whether paired text descriptions enable more controllable music-to-dance generation and whether our versatile method can seamlessly leverage both conditions. We conduct ablation studies on multimodal conditioning using our TMD dataset, comparing it with the single-condition setting using only music. The results in Table 8 demonstrate that introducing multimodal conditioning is more effective than using a single modality, and our model can effectively and adaptively handle multimodal conditions.

**Number of Layers.** To investigate the impact of our model by varying the number of layers $N$ in masked transformers, we conduct ablation studies on HumanML3D. Table 9 demonstrates the robustness of our model across different layer configurations.

## 4.5 QUALITATIVE EVALUATION

**Text-to-Motion Generation.** To qualitatively evaluate our performance in text-to-motion generation, we compare the visualizations generated by our method with those produced by previous state-of-the-art methods specializing in text-to-motion generation, including BAD (Hosseyni et al., 2024), BAMM (Pinyoanuntapong et al., 2024a), and MoMask (Guo et al., 2024). The text prompts

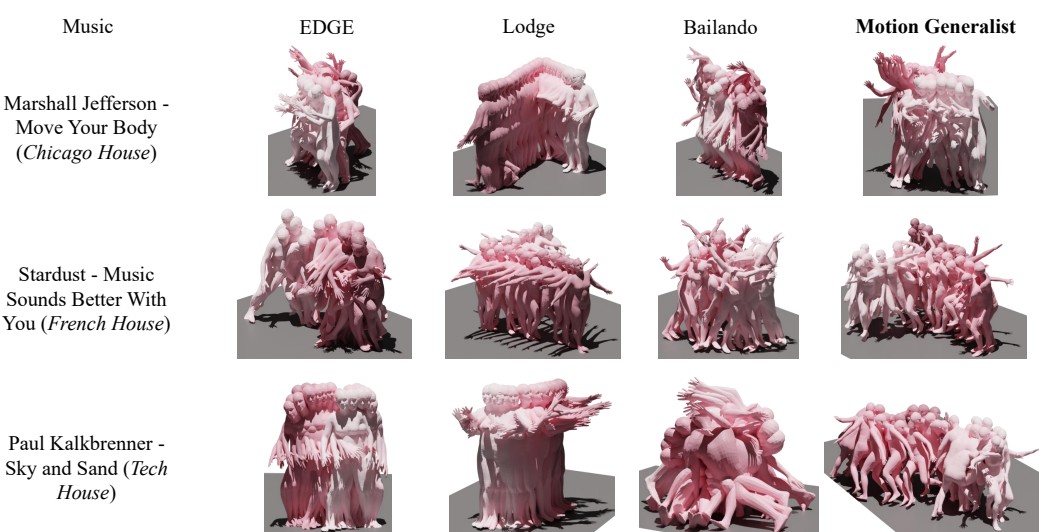

Figure 6: **Qualitative evaluation on music-to-dance generation.** We qualitatively compared the visualizations generated by our method with those produced by EDGE (Tseng et al., 2023), Lodge (Li et al., 2024b), and Bailando (Siyao et al., 2022).

Table 9: **Ablation study of number of layers** on HumanML3D (Guo et al., 2022a).

| Method | R Precision ↑ | | | FID↓ | MM Dist↓ | Diversity→ | MModality↑ |
|---|---|---|---|---|---|---|---|
| | Top 1 | Top 2 | Top 3 | | | | |
| Ground Truth | $0.511^{\pm.003}$ | $0.703^{\pm.003}$ | $0.797^{\pm.002}$ | $0.002^{\pm.000}$ | $2.974^{\pm.008}$ | $9.503^{\pm.065}$ | - |
| $N=2$ | $0.521^{\pm.006}$ | $0.725^{\pm.008}$ | $0.819^{\pm.005}$ | $0.079^{\pm.019}$ | $2.916^{\pm.033}$ | $9.598^{\pm.117}$ | $2.503^{\pm.024}$ |
| $N=4$ | $\mathbf{0.546}^{\pm.003}$ | $\mathbf{0.735}^{\pm.002}$ | $\mathbf{0.829}^{\pm.002}$ | $\mathbf{0.028}^{\pm.005}$ | $2.859^{\pm.010}$ | $9.521^{\pm.083}$ | $2.705^{\pm.068}$ |
| $N=6$ | $0.541^{\pm.007}$ | $0.733^{\pm.002}$ | $0.826^{\pm.010}$ | $0.029^{\pm.004}$ | $2.861^{\pm.010}$ | $\mathbf{9.517}^{\pm.094}$ | $2.673^{\pm.019}$ |
| $N=8$ | $0.544^{\pm.009}$ | $0.734^{\pm.003}$ | $0.826^{\pm.007}$ | $\mathbf{0.028}^{\pm.014}$ | $\mathbf{2.851}^{\pm.011}$ | $9.519^{\pm.057}$ | $\mathbf{2.711}^{\pm.032}$ |

are customized based on the HumanML3D (Guo et al., 2022a) test set. As shown in Figure 5 and video demos, our method generates motions with superior quality, greater diversity, and better alignment between text and motion compared to the previous state-of-the-art methods.

**Music-to-Dance Generation.** To evaluate the quality of our music-to-dance generation, we compare the dances generated by our method against those from state-of-the-art approaches, including EDGE (Tseng et al., 2023), Lodge (Li et al., 2024b), and Bailando (Siyao et al., 2022). Training on AIST++ (Li et al., 2021), we ensure the evaluation reflects diverse musical styles. As shown in Figure 6 and demonstrated in the accompanying videos, our method produces dances with better visual quality and achieves superior alignment with the beat and genre of the music, surpassing previous state-of-the-art techniques.

## 5 CONCLUSION

In conclusion, Motion Generalist presents a significant forward to motion generation by enabling adaptive and controllable multimodal conditioning. Our model introduces the attention-based masking within an autoregressive framework to focus on key frames and actions, addressing the challenge of prioritizing dynamic frames and body parts. Additionally, it bridges the gap in multimodal motion generation by aligning different input modalities both temporally and spatially, enhancing control and coherence. To further advance research in this area, we introduce the Text-Music-Dance (TMD) dataset, a pioneering benchmark with paired music and text. Extensive experiments demonstrate that our method outperforms prior approaches, achieving substantial improvements across multiple benchmarks. By tackling these challenges, Motion Generalist establishes a new paradigm for motion generation, offering a more versatile and precise framework for motion generation.

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

# APPENDIX

## A    LLM USE DECLARATION

Large Language Models (ChatGPT) were used exclusively to improve the clarity and fluency of English writing. They were not involved in research ideation, experimental design, data analysis, or interpretation. The authors take full responsibility for all content.

## B    ALGORITHMS

---

**Algorithm 1** Attention-based Masking

---

1: **Input:** Motion $\mathbf{M}$, Condition $\mathbf{C}$, Masking Ratio $\alpha$
2: **Define:** $\mathbf{T}$: Text space, $\mathbf{D}$: Audio space
3: **Step 1:** Define temporal $Q_{\text{temp}}, K_{\text{temp}}, V_{\text{temp}}$ and spatial $Q_{\text{spatial}}, K_{\text{spatial}}, V_{\text{spatial}}$
4: **if $\mathbf{C} \in \mathbf{T}$ then**
5:     $Q_{\text{temp}}, K_{\text{temp}}, V_{\text{temp}} \leftarrow (\mathbf{C}, \mathbf{M}), (\mathbf{C}, \mathbf{M}), (\mathbf{C}, \mathbf{M})$
6:     $Q_{\text{spatial}}, K_{\text{spatial}}, V_{\text{spatial}} \leftarrow \mathbf{C}, \mathbf{M}, \mathbf{M}$
7: **else if $\mathbf{C} \in \mathbf{D}$ or $\mathbf{C} \in \mathbf{T} \cap \mathbf{D}$ then**
8:     $Q_{\text{temp}}, K_{\text{temp}}, V_{\text{temp}} \leftarrow \mathbf{C}, \mathbf{M}, \mathbf{M}$
9:     $Q_{\text{spatial}}, K_{\text{spatial}}, V_{\text{spatial}} \leftarrow \mathbf{C}, \mathbf{M}, \mathbf{M}$
10: **end if**
11: **Step 2:** Compute Attention Scores
12: $A_{\text{temp}} = \text{AttentionScore}(Q_{\text{temp}}, K_{\text{temp}})$
13: $A_{\text{spatial}} = \text{AttentionScore}(Q_{\text{spatial}}, K_{\text{spatial}})$
14: **Step 3:** Apply Masking
15: Sort $A_{\text{temp}}$ and mask top $\alpha$ percent: $\text{mask}_{\text{temp}} = \{i \mid A_{\text{temp},i} \text{ in top } \alpha\%\}$
16: Sort $A_{\text{spatial}}$ and mask top $\alpha$ percent: $\text{mask}_{\text{spatial}} = \{i \mid A_{\text{spatial},i} \text{ in top } \alpha\%\}$
17: **Output:** Masked motion sequence $\mathbf{M}_{masked}$

---

**Algorithm 2** Temporal Adaptive Transformer

---

1: **Input:** Motion $\mathbf{M}$, Condition $\mathbf{C}$, Mask Ratio $\alpha$
2: **Define:** $\mathbf{T}$: Text space, $\mathbf{A}$: Audio space
3: **Step 1:** Apply Attention-based Masking to $\mathbf{M}$
4: $\mathbf{M}_{\text{masked}} \leftarrow \text{Attention-based Masking}(\mathbf{M}, \mathbf{C}, \alpha)$
5: **Step 2:** Define Q, K, V
6: **if $\mathbf{C} \in \mathbf{T}$ then**
7:     $Q, K, V \leftarrow (\mathbf{C}, \mathbf{M}_{\text{masked}}), (\mathbf{C}, \mathbf{M}_{\text{masked}}), (\mathbf{C}, \mathbf{M}_{\text{masked}})$
8: **else if $\mathbf{C} \in \mathbf{A}$ or $\mathbf{C} \in \mathbf{T} \cap \mathbf{A}$ then**
9:     $Q, K, V \leftarrow \mathbf{M}_{\text{masked}}, \mathbf{C}, \mathbf{C}$
10: **end if**
11: **Step 3:** Compute Temporal Attention
12: $\mathbf{M}_{\text{restored}} = \text{Attention}(Q, K, V)$
13: **Output:** Restored Motion Sequence $\mathbf{M}_{\text{restored}}$

---

**Algorithm 3** Spatial Aligning Transformer

---

1: **Input:** Motion $\mathbf{M}'$, Condition $\mathbf{C}'$, Mask Ratio $\alpha$
2: **Step 1:** Apply Attention-based Masking to $\mathbf{M}'$
3: $\mathbf{M}'_{\text{masked}} \leftarrow \text{Attention-based Masking}(\mathbf{M}', \mathbf{C}', \alpha)$
4: **Step 2:** Define Q, K, V
5: $Q, K, V \leftarrow \mathbf{M}'_{\text{masked}}, \mathbf{C}', \mathbf{C}'$
6: **Step 3:** Compute Spatial Attention
7: $\mathbf{M}'_{\text{restored}} = \text{Attention}(Q, K, V)$
8: **Output:** Restored Motion Sequence $\mathbf{M}'_{\text{restored}}$

---

Table 10: **Comprehensive comparison on HumanML3D (Guo et al., 2022a) and KIT-ML (Plappert et al., 2016).** The best and runner-up values are **bold** and underlined. The right arrow → indicates that closer values to ground truth are better. Multimodal motion generation methods are highlighted in blue.

| Datasets | Method | R Precision ↑ | | | FID↓ | MultiModal Dist↓ | Diversity→ | MultiModality↑ |
|---|---|---|---|---|---|---|---|---|
| | | Top 1 | Top 2 | Top 3 | | | | |
| | Ground Truth | $0.511^{\pm.003}$ | $0.703^{\pm.003}$ | $0.797^{\pm.002}$ | $0.002^{\pm.000}$ | $2.974^{\pm.008}$ | $9.503^{\pm.065}$ | - |
| | TEMOS (Petrovich et al., 2022) | $0.424^{\pm.002}$ | $0.612^{\pm.002}$ | $0.722^{\pm.002}$ | $3.734^{\pm.028}$ | $3.703^{\pm.008}$ | $8.973^{\pm.071}$ | $0.368^{\pm.018}$ |
| | TM2T (Guo et al., 2022b) | $0.424^{\pm.003}$ | $0.618^{\pm.003}$ | $0.729^{\pm.002}$ | $1.501^{\pm.017}$ | $3.467^{\pm.011}$ | $8.589^{\pm.076}$ | $2.424^{\pm.093}$ |
| | T2M (Guo et al., 2022a) | $0.457^{\pm.002}$ | $0.639^{\pm.003}$ | $0.740^{\pm.003}$ | $1.067^{\pm.002}$ | $3.340^{\pm.008}$ | $9.188^{\pm.002}$ | $2.090^{\pm.083}$ |
| | TM2D (Gong et al., 2023) | $0.319^{\pm.000}$ | - | - | $1.021^{\pm.000}$ | $4.098^{\pm.000}$ | $9.513^{\pm.000}$ | $4.139^{\pm.000}$ |
| | MotionGPT (Zhang et al.) (Zhang et al., 2024f) | $0.364^{\pm.005}$ | $0.533^{\pm.003}$ | $0.629^{\pm.004}$ | $0.805^{\pm.002}$ | $3.914^{\pm.013}$ | $9.972^{\pm.026}$ | $2.473^{\pm.041}$ |
| | MotionDiffuse (Zhang et al., 2024d) | $0.491^{\pm.001}$ | $0.681^{\pm.001}$ | $0.782^{\pm.001}$ | $0.630^{\pm.001}$ | $3.113^{\pm.001}$ | $9.410^{\pm.049}$ | $1.553^{\pm.042}$ |
| | MDM (Tevet et al., 2022) | $0.320^{\pm.005}$ | $0.498^{\pm.004}$ | $0.611^{\pm.007}$ | $0.544^{\pm.044}$ | $5.566^{\pm.027}$ | $9.559^{\pm.086}$ | $2.799^{\pm.072}$ |
| | MotionLLM (Wu et al., 2024) | $0.482^{\pm.004}$ | $0.672^{\pm.003}$ | $0.770^{\pm.002}$ | $0.491^{\pm.019}$ | $3.138^{\pm.010}$ | $9.838^{\pm.244}$ | - |
| | MLD (Chen et al., 2023) | $0.481^{\pm.003}$ | $0.673^{\pm.003}$ | $0.772^{\pm.002}$ | $0.473^{\pm.013}$ | $3.196^{\pm.010}$ | $9.724^{\pm.082}$ | $2.413^{\pm.079}$ |
| | M2DM (Kong et al., 2023) | $0.497^{\pm.003}$ | $0.682^{\pm.002}$ | $0.763^{\pm.002}$ | $0.352^{\pm.005}$ | $3.134^{\pm.010}$ | $9.926^{\pm.073}$ | $3.587^{\pm.072}$ |
| | MotionLCM (Dai et al., 2024) | $0.502^{\pm.003}$ | $0.698^{\pm.002}$ | $0.798^{\pm.002}$ | $0.304^{\pm.012}$ | $3.012^{\pm.007}$ | $9.607^{\pm.066}$ | $2.259^{\pm.092}$ |
| | Motion Mamba (Zhang et al., 2025) | $0.502^{\pm.003}$ | $0.693^{\pm.002}$ | $0.792^{\pm.002}$ | $0.281^{\pm.009}$ | $3.060^{\pm.058}$ | $9.871^{\pm.084}$ | $2.294^{\pm.058}$ |
| Human ML3D (Guo et al., 2022a) | Fg-T2M (Wang et al., 2023b) | $0.492^{\pm.002}$ | $0.683^{\pm.003}$ | $0.783^{\pm.002}$ | $0.243^{\pm.019}$ | $3.109^{\pm.007}$ | $9.278^{\pm.072}$ | $1.614^{\pm.049}$ |
| | MotionGPT (Jiang et al.) (Jiang et al., 2023a) | $0.492^{\pm.003}$ | $0.681^{\pm.003}$ | $0.778^{\pm.002}$ | $0.232^{\pm.008}$ | $3.096^{\pm.008}$ | $9.528^{\pm.071}$ | $2.008^{\pm.084}$ |
| | MotionGPT-2 (Wang et al., 2024b) | $0.496^{\pm.002}$ | $0.691^{\pm.003}$ | $0.782^{\pm.004}$ | $0.191^{\pm.004}$ | $3.080^{\pm.013}$ | $9.860^{\pm.026}$ | $2.137^{\pm.022}$ |
| | MotionCraft (Bian et al., 2024) | $0.501^{\pm.003}$ | $0.697^{\pm.003}$ | $0.796^{\pm.002}$ | $0.173^{\pm.002}$ | $3.025^{\pm.008}$ | $9.543^{\pm.098}$ | - |
| | FineMoGen (Zhang et al., 2023c) | $0.504^{\pm.002}$ | $0.690^{\pm.002}$ | $0.784^{\pm.002}$ | $0.151^{\pm.008}$ | $2.998^{\pm.008}$ | $9.263^{\pm.094}$ | $2.696^{\pm.079}$ |
| | T2M-GPT (Zhang et al., 2023a) | $0.492^{\pm.003}$ | $0.679^{\pm.002}$ | $0.775^{\pm.002}$ | $0.141^{\pm.005}$ | $3.121^{\pm.009}$ | $9.722^{\pm.082}$ | $1.831^{\pm.048}$ |
| | GraphMotion (Jin et al., 2024) | $0.504^{\pm.003}$ | $0.699^{\pm.002}$ | $0.785^{\pm.002}$ | $0.116^{\pm.007}$ | $3.070^{\pm.008}$ | $9.692^{\pm.067}$ | $2.766^{\pm.096}$ |
| | EMDM (Zhou et al., 2023a) | $0.498^{\pm.007}$ | $0.684^{\pm.006}$ | $0.786^{\pm.006}$ | $0.112^{\pm.019}$ | $3.110^{\pm.027}$ | $9.551^{\pm.078}$ | $1.641^{\pm.078}$ |
| | AttT2M (Zhong et al., 2023) | $0.499^{\pm.003}$ | $0.690^{\pm.002}$ | $0.786^{\pm.002}$ | $0.112^{\pm.006}$ | $3.038^{\pm.007}$ | $9.700^{\pm.090}$ | $2.452^{\pm.051}$ |
| | GUESS (Gao et al., 2024) | $0.503^{\pm.003}$ | $0.688^{\pm.002}$ | $0.787^{\pm.002}$ | $0.109^{\pm.007}$ | $3.006^{\pm.007}$ | $9.826^{\pm.104}$ | $2.430^{\pm.100}$ |
| | ParCo (Zou et al., 2024) | $0.515^{\pm.003}$ | $0.706^{\pm.003}$ | $0.801^{\pm.002}$ | $0.109^{\pm.005}$ | $2.927^{\pm.008}$ | $9.576^{\pm.088}$ | $1.382^{\pm.060}$ |
| | ReMoDiffuse (Zhang et al., 2023b) | $0.510^{\pm.005}$ | $0.698^{\pm.006}$ | $0.795^{\pm.004}$ | $0.103^{\pm.004}$ | $2.974^{\pm.016}$ | $9.018^{\pm.075}$ | $1.795^{\pm.043}$ |
| | MotionCLR (Chen et al., 2024a) | $0.542^{\pm.001}$ | $0.733^{\pm.002}$ | $0.827^{\pm.002}$ | $0.099^{\pm.003}$ | $2.981^{\pm.011}$ | - | $2.145^{\pm.043}$ |
| | StableMoFusion (Huang et al., 2024a) | $\mathbf{0.553^{\pm.003}}$ | $\mathbf{0.748^{\pm.002}}$ | $\mathbf{0.841^{\pm.002}}$ | $0.098^{\pm.003}$ | - | $9.748^{\pm.092}$ | $1.774^{\pm.051}$ |
| | MMM (Pinyoanuntapong et al., 2024b) | $0.504^{\pm.003}$ | $0.696^{\pm.003}$ | $0.794^{\pm.002}$ | $0.080^{\pm.003}$ | $2.998^{\pm.007}$ | $9.411^{\pm.058}$ | $1.164^{\pm.041}$ |
| | DiverseMotion (Lou et al., 2023) | $0.515^{\pm.003}$ | $0.706^{\pm.002}$ | $0.802^{\pm.002}$ | $0.072^{\pm.004}$ | $2.941^{\pm.007}$ | $9.683^{\pm.102}$ | $1.869^{\pm.089}$ |
| | BAD (Hosseyni et al., 2024) | $0.517^{\pm.002}$ | $0.713^{\pm.003}$ | $0.808^{\pm.003}$ | $0.065^{\pm.003}$ | $2.901^{\pm.008}$ | $9.694^{\pm.068}$ | $1.194^{\pm.044}$ |
| | BAMM (Pinyoanuntapong et al., 2024a) | $0.525^{\pm.002}$ | $0.720^{\pm.003}$ | $0.814^{\pm.003}$ | $0.055^{\pm.002}$ | $2.919^{\pm.008}$ | $9.717^{\pm.089}$ | $1.687^{\pm.051}$ |
| | MCM (Ling et al., 2024) | $0.502^{\pm.002}$ | $0.692^{\pm.004}$ | $0.788^{\pm.006}$ | $0.053^{\pm.007}$ | $3.037^{\pm.003}$ | $9.585^{\pm.082}$ | $0.810^{\pm.023}$ |
| | MoMask (Guo et al., 2024) | $0.521^{\pm.002}$ | $0.713^{\pm.002}$ | $0.807^{\pm.002}$ | $0.045^{\pm.002}$ | $2.958^{\pm.008}$ | - | $1.241^{\pm.040}$ |
| | LMM (Zhang et al., 2024e) | $0.525^{\pm.002}$ | $0.719^{\pm.002}$ | $0.811^{\pm.002}$ | $0.040^{\pm.002}$ | $2.943^{\pm.012}$ | $9.814^{\pm.076}$ | $2.683^{\pm.054}$ |
| | MoGenTS (Yuan et al., 2024) | $0.529^{\pm.003}$ | $0.719^{\pm.002}$ | $0.812^{\pm.002}$ | $0.033^{\pm.001}$ | $\underline{2.867^{\pm.006}}$ | $9.570^{\pm.077}$ | - |
| | **Motion Generalist (Ours)** | $\underline{0.546^{\pm.003}}$ | $\underline{0.735^{\pm.002}}$ | $\underline{0.829^{\pm.002}}$ | $\mathbf{0.028^{\pm.005}}$ | $\mathbf{2.859^{\pm.010}}$ | $\underline{9.521^{\pm.083}}$ | $2.705^{\pm.068}$ |
| | Ground Truth | $0.424^{\pm.005}$ | $0.649^{\pm.006}$ | $0.779^{\pm.006}$ | $0.031^{\pm.004}$ | $2.788^{\pm.012}$ | $11.08^{\pm.097}$ | - |
| | TEMOS (Petrovich et al., 2022) | $0.353^{\pm.006}$ | $0.561^{\pm.007}$ | $0.687^{\pm.005}$ | $3.717^{\pm.051}$ | $3.417^{\pm.019}$ | $10.84^{\pm.100}$ | $0.532^{\pm.034}$ |
| | TM2T (Guo et al., 2022b) | $0.280^{\pm.005}$ | $0.463^{\pm.006}$ | $0.587^{\pm.005}$ | $3.599^{\pm.153}$ | $4.591^{\pm.026}$ | $9.473^{\pm.117}$ | $3.292^{\pm.081}$ |
| | T2M (Guo et al., 2022a) | $0.370^{\pm.005}$ | $0.569^{\pm.007}$ | $0.693^{\pm.007}$ | $2.770^{\pm.109}$ | $3.401^{\pm.008}$ | $10.91^{\pm.119}$ | $1.482^{\pm.065}$ |
| | MotionDiffuse (Zhang et al., 2024d) | $0.417^{\pm.004}$ | $0.621^{\pm.004}$ | $0.739^{\pm.004}$ | $1.954^{\pm.062}$ | $2.958^{\pm.005}$ | $11.10^{\pm.143}$ | $0.753^{\pm.013}$ |
| | MDM (Tevet et al., 2022) | $0.164^{\pm.004}$ | $0.291^{\pm.004}$ | $0.396^{\pm.004}$ | $0.497^{\pm.021}$ | $9.190^{\pm.022}$ | $10.85^{\pm.109}$ | $1.907^{\pm.214}$ |
| | MLD (Chen et al., 2023) | $0.390^{\pm.003}$ | $0.609^{\pm.003}$ | $0.734^{\pm.002}$ | $0.404^{\pm.013}$ | $3.204^{\pm.010}$ | $10.80^{\pm.082}$ | $2.192^{\pm.079}$ |
| | M2DM (Kong et al., 2023) | $0.405^{\pm.003}$ | $0.629^{\pm.005}$ | $0.739^{\pm.004}$ | $0.502^{\pm.049}$ | $3.012^{\pm.015}$ | $11.38^{\pm.079}$ | $\mathbf{3.273^{\pm.045}}$ |
| | Motion Mamba (Zhang et al., 2025) | $0.419^{\pm.006}$ | $0.645^{\pm.005}$ | $0.765^{\pm.006}$ | $0.307^{\pm.041}$ | $3.021^{\pm.025}$ | $11.02^{\pm.098}$ | $1.678^{\pm.064}$ |
| | Fg-T2M (Wang et al., 2023b) | $0.418^{\pm.005}$ | $0.626^{\pm.004}$ | $0.745^{\pm.004}$ | $0.571^{\pm.047}$ | $3.114^{\pm.015}$ | $10.93^{\pm.083}$ | $1.019^{\pm.029}$ |
| | MotionGPT (Zhang et al.) (Zhang et al., 2024f) | $0.340^{\pm.002}$ | $0.570^{\pm.003}$ | $0.660^{\pm.004}$ | $0.868^{\pm.032}$ | $3.721^{\pm.018}$ | $9.972^{\pm.026}$ | $2.296^{\pm.022}$ |
| KIT-ML (Plappert et al., 2016) | MotionGPT (Jiang et al.) (Jiang et al., 2023a) | $0.366^{\pm.005}$ | $0.558^{\pm.004}$ | $0.680^{\pm.005}$ | $0.510^{\pm.016}$ | $3.527^{\pm.021}$ | $10.35^{\pm.084}$ | $2.328^{\pm.117}$ |
| | MotionGPT-2 (Wang et al., 2024b) | $0.427^{\pm.003}$ | $0.627^{\pm.002}$ | $0.764^{\pm.003}$ | $0.614^{\pm.005}$ | $3.164^{\pm.013}$ | $11.26^{\pm.026}$ | $2.357^{\pm.022}$ |
| | FineMoGen (Zhang et al., 2023c) | $0.432^{\pm.006}$ | $0.649^{\pm.005}$ | $0.772^{\pm.006}$ | $0.178^{\pm.007}$ | $2.869^{\pm.014}$ | $10.85^{\pm.115}$ | $1.877^{\pm.093}$ |
| | T2M-GPT (Zhang et al., 2023a) | $0.416^{\pm.006}$ | $0.627^{\pm.006}$ | $0.745^{\pm.006}$ | $0.514^{\pm.029}$ | $3.007^{\pm.023}$ | $10.92^{\pm.108}$ | $1.570^{\pm.039}$ |
| | GraphMotion (Jin et al., 2024) | $0.417^{\pm.008}$ | $0.635^{\pm.006}$ | $0.755^{\pm.004}$ | $0.262^{\pm.021}$ | $3.085^{\pm.031}$ | $11.21^{\pm.106}$ | $\mathbf{3.568^{\pm.132}}$ |
| | EMDM (Zhou et al., 2023a) | $0.443^{\pm.006}$ | $0.660^{\pm.006}$ | $0.780^{\pm.005}$ | $0.261^{\pm.014}$ | $2.874^{\pm.015}$ | $10.96^{\pm.093}$ | $1.343^{\pm.089}$ |
| | AttT2M (Zhong et al., 2023) | $0.413^{\pm.006}$ | $0.632^{\pm.006}$ | $0.751^{\pm.006}$ | $0.870^{\pm.039}$ | $3.039^{\pm.021}$ | $10.96^{\pm.123}$ | $2.281^{\pm.047}$ |
| | GUESS (Gao et al., 2024) | $0.425^{\pm.005}$ | $0.632^{\pm.007}$ | $0.751^{\pm.005}$ | $0.371^{\pm.020}$ | $2.421^{\pm.022}$ | $10.93^{\pm.110}$ | $2.732^{\pm.084}$ |
| | ParCo (Zou et al., 2024) | $0.430^{\pm.004}$ | $0.649^{\pm.007}$ | $0.772^{\pm.006}$ | $0.453^{\pm.027}$ | $2.820^{\pm.028}$ | $10.95^{\pm.094}$ | $1.245^{\pm.022}$ |
| | ReMoDiffuse (Zhang et al., 2023b) | $0.427^{\pm.014}$ | $0.641^{\pm.004}$ | $0.765^{\pm.055}$ | $0.155^{\pm.006}$ | $2.814^{\pm.012}$ | $10.80^{\pm.105}$ | $1.239^{\pm.028}$ |
| | StableMoFusion (Huang et al., 2024a) | $\underline{0.445^{\pm.006}}$ | $0.660^{\pm.005}$ | $0.782^{\pm.004}$ | $0.258^{\pm.029}$ | - | $10.94^{\pm.077}$ | $1.362^{\pm.062}$ |
| | MMM (Pinyoanuntapong et al., 2024b) | $0.404^{\pm.005}$ | $0.621^{\pm.005}$ | $0.744^{\pm.004}$ | $0.316^{\pm.028}$ | $2.977^{\pm.019}$ | $10.91^{\pm.101}$ | $1.232^{\pm.039}$ |
| | DiverseMotion (Lou et al., 2023) | $0.416^{\pm.006}$ | $0.637^{\pm.008}$ | $0.760^{\pm.011}$ | $0.468^{\pm.098}$ | $2.892^{\pm.041}$ | $10.87^{\pm.101}$ | $2.062^{\pm.079}$ |
| | BAD (Hosseyni et al., 2024) | $0.417^{\pm.006}$ | $0.631^{\pm.006}$ | $0.750^{\pm.006}$ | $0.221^{\pm.012}$ | $2.941^{\pm.005}$ | $\mathbf{11.00^{\pm.100}}$ | $1.170^{\pm.047}$ |
| | BAMM (Pinyoanuntapong et al., 2024a) | $0.438^{\pm.009}$ | $0.661^{\pm.009}$ | $0.788^{\pm.005}$ | $0.183^{\pm.013}$ | $2.723^{\pm.026}$ | $\underline{11.01^{\pm.094}}$ | $1.609^{\pm.065}$ |
| | MoMask (Guo et al., 2024) | $0.433^{\pm.007}$ | $0.656^{\pm.005}$ | $0.781^{\pm.005}$ | $0.204^{\pm.011}$ | $2.779^{\pm.022}$ | - | $1.131^{\pm.043}$ |
| | LMM (Zhang et al., 2024e) | $0.430^{\pm.015}$ | $0.653^{\pm.017}$ | $0.779^{\pm.014}$ | $\underline{0.137^{\pm.023}}$ | $2.791^{\pm.018}$ | $11.24^{\pm.103}$ | $1.885^{\pm.127}$ |
| | MoGenTS (Yuan et al., 2024) | $\underline{0.445^{\pm.006}}$ | $\underline{0.671^{\pm.006}}$ | $\underline{0.797^{\pm.005}}$ | $0.143^{\pm.004}$ | $\underline{2.711^{\pm.024}}$ | $10.92^{\pm.090}$ | - |
| | **Motion Generalist (Ours)** | $\mathbf{0.449^{\pm.007}}$ | $\mathbf{0.678^{\pm.004}}$ | $\mathbf{0.802^{\pm.006}}$ | $\mathbf{0.131^{\pm.003}}$ | $\mathbf{2.705^{\pm.024}}$ | $10.94^{\pm.098}$ | $1.374^{\pm.069}$ |

## C FULL COMPARISON FOR TEXT-TO-MOTION

To comprehensively showcase our method's performance in text-to-motion generation, we present a full comparison with previous T2M approaches in Table 10. The results demonstrate that our method consistently outperforms others, achieving state-of-the-art performance on both HumanML3D (Guo et al., 2022a) and KIT-ML (Plappert et al., 2016) datasets.

## D USER STUDY

This study provides a comprehensive evaluation of our motion generation. We assessed the real-world applicability of four motion videos generated by Motion Generalist and baseline models, as evaluated by 50 participants through a Google Forms survey. Figure 7 displays the User Interface (UI) used in our user study, showcasing 3–4 videos (Video 1 to 3/4), each featuring distinct motion animations from the same model, and four videos for comparing different models (Video A to D).

Participants evaluate these animations based on aspects such as motion accuracy and overall user experience. They rate each aspect from 1 (low) to 3 (high) to assess how well the animations mirror real-world movements and their engagement level. In the comparison section, participants select the model with the best performance. This evaluation aims to determine the realism and engagement effectiveness of each motion. The evaluation consisted of three groups of motions: text-to-motion, music-to-dance, and text-and-music-to-dance. The results are as follows:

**Text-to-Motion:**

- Our motion quality rating is **2.84**. Additionally, **86%** of participants believe that our method demonstrates high-quality motion generation with minimal jitter, sliding, and unrealistic movements.
- Our motion diversity rating is **2.68**. Additionally, **72%** of participants believe that our method generates complex and diverse motions.
- Our text-motion alignment rating is **2.74**. Additionally, **76%** of participants believe that our method generates motion that is well-aligned with their text condition.
- **88%** of participants believe that our method outperforms other methods.

**Music-to-Dance:**

- Our dance quality rating is **2.82**. Additionally, **82%** of participants believe that our method demonstrates high-quality dance generation with minimal jitter, sliding, and unrealistic movements.
- Our dance diversity rating is **2.88**. Additionally, **88%** of participants believe that our method generates complex and diverse dance.
- Our music-dance alignment rating is **2.66**. Additionally, **70%** of participants believe that our method generates dance that is well-aligned with the music genre and beats.
- **74%** of participants believe that our method outperforms other methods.

**Text and Music to Dance:**

- Our dance quality rating is **2.74**. Additionally, **76%** of participants believe that our method demonstrates high-quality dance generation with minimal jitter, sliding, and unrealistic movements.
- Our multimodal condition rating is **2.88**. Additionally, **88%** of participants believe that text enhances music's conditioning in dance generation.
- Our text&music-dance aligment rating is **2.76**. There are **80%** participants believe that our method generates dance that is well-aligned with the both music and text.
- **78%** of participants believe that our method outperforms other methods.

## E MODEL EFFICIENCY

We evaluate the inference efficiency of motion generation compared to various methods. The inference cost is calculated as the average inference time over 100 samples on one NVIDIA GeForce RTX 2080 Ti device. Compared to baseline methods, Motion Generalist achieves an outstanding balance between generation quality and efficiency, as shown in Figure 8.

## F APPLICATION: 4D AVATAR GENERATION

One of the most significant applications for conditional motion generation is 4D avatar generation. Previous methods (Li et al., 2024c; Ren et al., 2023; Wu et al., 2025; Yin et al., 2023; Jiang et al., 2023b; Ren et al., 2024; Zeng et al.,

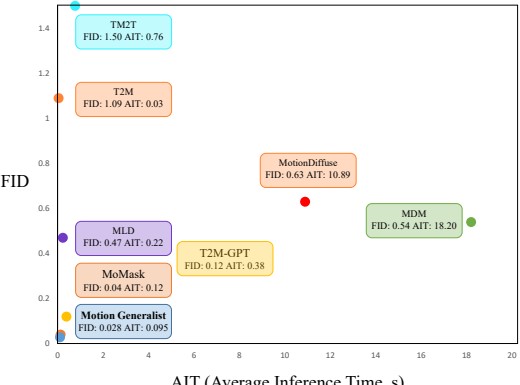

Figure 8: **Comparisons on FID and AIT.** All tests are conducted on the same NVIDIA GeForce RTX 2080 Ti. The closer the model is to the origin, the better.

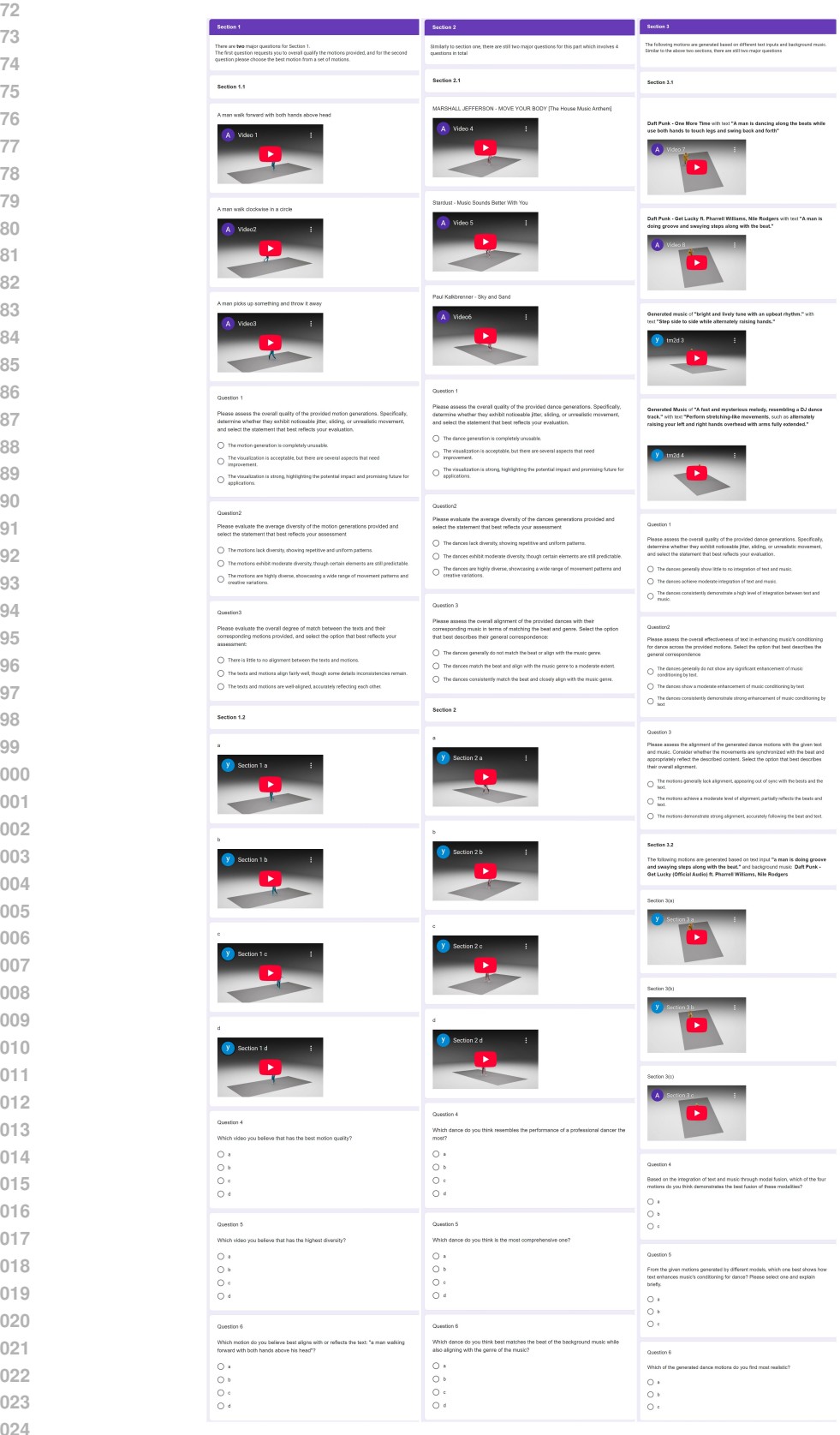

Figure 7: **User study form.** The User Interface (UI) used in our user study.

2025; Wang et al., 2024a; Hu et al., 2024; Chen et al., 2024b) for 4D avatar generation reconstruct dynamic avatars using 4D Gaussians to model both spatial and temporal dimensions from video data. Other approaches (Zhang et al., 2024b; Zheng et al., 2024; Bahmani et al., 2024; Zhao et al., 2023) integrate video diffusion (Wang et al., 2023a; Blattmann et al., 2023) with geometry-aware diffusion models (Shi et al., 2023) to ensure spatial consistency and achieve a visually appealing appearance. However, these methods face two significant challenges: (1) limited diversity and control over motion (Li et al., 2024c; Wu et al., 2025; Jiang et al., 2023b), and (2) inconsistencies in the mesh appearance at different time points (Ren et al., 2023).

Therefore, we propose a comprehensive feed-forward approach for 4D avatar generation with only a single prompt, leveraging both Motion Generalist and off-the-shelf tools, as illustrated in Figure 10. A single prompt serves as the input, when music is not provided as a condition, Stable Audio Open (Evans et al., 2024) generates background music as the audio condition. The text and audio conditions are then processed by the multimodal motion generator to create a high-quality motion sequence. Simultaneously, 3D avatar generation employs Tripo AI 2.0 (tri, 2024) to produce candidate meshes, as shown in Figure 9, and the Selective Rigging Mechanism (SRM) identifies the optimal rigged mesh. The motion sequence is then retargeted to the avatar meshes, resulting in a complete 4D avatar.

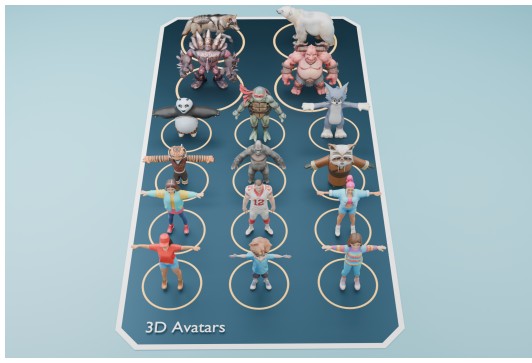

Figure 9: **3D Avatars.** This figure shows examples of 3D avatars generated by Tripo AI 2.0 (tri, 2024). These avatars will later serve as candidates for our Selective Rigging Mechanism.

As shown in the Figure 10, automatic rigging is crucial for 4D generation, directly affecting the precision and realism of avatar movements (Zhang et al., 2024c;i). Although numerous optimization-based approaches (Baran & Popović, 2007; Feng et al., 2015; Borosán et al., 2012; Pantuwong & Sugimoto, 2011) have been proposed to achieve fully automated rigging, the outcomes are often unsatisfactory due to the diverse appearances of meshes. Hence, achieving high-quality automatic rigging has become an important challenge to address in skeleton-based avatar generation. To improve the underperformance of automatic rigging in generated avatars, we introduced a straightforward yet effective **Selective Rigging Mechanism** that selects the best-rigged 3D avatar from multiple candidates, enhancing the realism of the avatar's motion visualization.

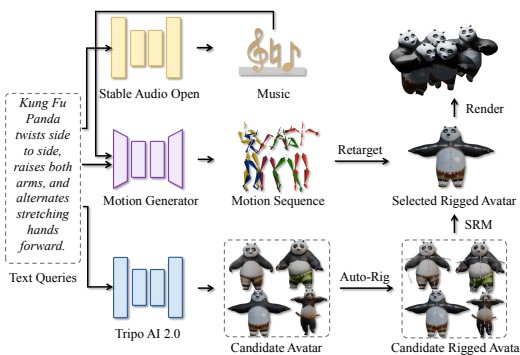

Figure 10: **4D Avatar Generation.** This approach enables 4D avatar generation conditioned on multimodal inputs, achievable with just a single text prompt.

### F.1    SELECTIVE RIGGING MECHANISM

To improve automatic rigging performance and reduce the need for human-in-the-loop adjustments, the Selective Rigging Mechanism (SRM) presents a two-stage selection process with constraints. This mechanism identifies the optimal rigging from a set of candidate avatars, as shown in Figure 10.

**Stage 1: Centroid-Based Filtering**. The purpose of this stage is to identify point clouds with centroids positioned within a balanced and plausible bounding region for rigging. Each animated point cloud is defined by a set of 3D coordinates $\mathbf{P} = \{\mathbf{p}_1, \mathbf{p}_2, \ldots, \mathbf{p}_N\}$, where $\mathbf{p}_i = (x_i, y_i, z_i) \in \mathbb{R}^3$, representing the character's surface. The centroid $\mathbf{G}_{\text{cloud}}$ of the point cloud, serving as an approxi-

mate center of mass, is calculated as

$$\mathbf{G}_{\text{cloud}} = \left( \frac{1}{N} \sum_{i=1}^{N} x_i, \frac{1}{N} \sum_{i=1}^{N} y_i, \frac{1}{N} \sum_{i=1}^{N} z_i \right).$$

The bounding box and stability filters ensure that $\mathbf{G}_{\text{cloud}}$ falls within a spatial region aligned with the character's rigging needs. Specifically, the bounding box constraint requires $-1 < X_G < 1$, $-1 < Y_G < 1$, $-1 < Z_G < 1$, while the stability constraint approximates balance by enforcing $|X_G| \approx 0$, $|Y_G| \approx 0$, and $Z_G > 0$.

These constraints are physically motivated: (1) Minimal lateral displacement, represented by $|X_G| \approx 0$ and $|Y_G| \approx 0$, keeps the center of mass near the $z$-axis, avoiding lateral imbalances. (2) Ensuring $Z_G > 0$ places the centroid above the ground plane, maintaining a logical upright character orientation.

**Stage 2: Joint Weight Optimization**. This stage's goal is to select the point cloud configuration with the best joint weight distribution to support stable, smooth deformation during animation. Each vertex $i$ has joint weights $w_{i1}, w_{i2}, \ldots, w_{in}$, where each $w_{ij}$ denotes the influence of joint $j$ on vertex $i$. To achieve realistic deformation, these weights must sum to 1 across all joints for each vertex, as specified by (Le & Deng, 2012). The weight normalization condition is expressed as:

$$\sum_{j=1}^{n} w_{ij} = 1, \quad \forall i \in \{1, 2, \ldots, N\}$$

where $N$ is the total number of points in $\mathbf{P}$.

To evaluate and select the optimal point cloud from $M$ candidates, we define a loss function based on the average deviation from the ideal weight sum. For each point cloud, we calculate the average weight sum $S$ as

$$S = \frac{1}{N} \sum_{i=1}^{N} \sum_{j=1}^{n} w_{ij}.$$

Our loss function, defined as the absolute difference $|S - 1|$, quantifies how close each point cloud's weight distribution is to the ideal configuration. Minimizing $|S - 1|$ allows us to select the point cloud whose joint weights best satisfy the normalization condition. Thus, the optimal point cloud $\mathbf{P}_{\text{optimal}}$ is chosen as:

$$\mathbf{P}_{\text{optimal}} = \arg \min_{\mathbf{P}_k} |S_k - 1|,$$

where $k \in \{1, 2, \ldots, M\}$, $\mathbf{P}_k$ is the $k$-th candidate point cloud, and $S_k$ is the average weight sum for $\mathbf{P}_k$. By selecting the configuration with the smallest $|S - 1|$, we ensure a joint weight distribution close to ideal, supporting stable, natural rigging and deformation in animation.

To showcase our improvement and efficiency in rigging with SRM on TMD dataset, we create 200 avatar descriptions along with text randomly selected from TMD to generate corresponding avatars using TripoAI 2.0 (tri, 2024). We then evaluate our SRM with different candidate numbers $k$ based on the average weighted sum $S$ in terms of rigging quality, where a value of $S$ closer to 1 indicates better performance. The results show in in Table 11. Video demos of the generated 4D avatars can also be found in the supplementary materials.

Table 11: SRM evaluation.

| Method | $S \to 1$ | AIT(s) $\downarrow$ |
|---|---|---|
| MagicPose4D | 1.93 | 0.138 |
| SRM ($k = 1$) | 1.78 | 0.094 |
| SRM ($k = 3$) | 1.36 | 0.105 |
| SRM ($k = 5$) | 1.06 | 0.117 |

# G    QUALITATIVE EVALUATION

In the main text, we have already demonstrated the qualitative evaluation of *text-to-motion* generation and *music-to-dance* generation. Here, we showcase some examples of the visualization of *text-and-music-to-dance* generation.

---

**Algorithm 4** Selective Rigging Mechanism

---

1: **Input:** $M$ point clouds $\{\mathbf{P}_1, \ldots, \mathbf{P}_M\}$, each with $N$ 3D vertices and joint weights $\{w_{i1}, \ldots, w_{in}\}$ for each vertex $i$
2: **Stage 1: Centroid Filtering**
3: **for** each $\mathbf{P}_k$ **do**
4:    Compute centroid $\mathbf{c}_k = (X_k, Y_k, Z_k)$
5:    **if** $\mathbf{c}_k$ is outside $-1 < X_k, Y_k, Z_k < 1$ or not close to $(0, 0, Z > 0)$ **then**
6:       Discard $\mathbf{P}_k$
7:    **end if**
8: **end for**
9: **Stage 2: Weight Optimization**
10: **for** each remaining $\mathbf{P}_k$ **do**
11:    Calculate $S_k = \frac{1}{N} \sum_{i=1}^{N} \sum_{j=1}^{n} w_{ij}$
12:    Compute deviation $\Delta_k = |S_k - 1|$
13: **end for**
14: Select $\mathbf{P}_{\text{optimal}} = \arg\min_{\mathbf{P}_k} \Delta_k$
15: **Output:** Optimal point cloud $\mathbf{P}_{\text{optimal}}$

---

Figure 11: **Qualitative evaluation on text-&-music-to-dance generation.** We qualitatively compared the visualizations generated by our method with those produced by TM2D (Gong et al., 2023) and MotionCraft (Bian et al., 2024).

## G.1 TEXT-&-MUSIC-TO-DANCE

To evaluate the quality of our text-and-music-to-dance generation, we compare the dances generated by our method against those from open-source state-of-the-art multi-task models, including TM2D (Gong et al., 2023) and MotionCraft (Bian et al., 2024). Similar to Section 4.3 in the main text,

despite these multi-task models lacking the ability to simultaneously take two modalities of conditions, we combined their condition embeddings and compared them with our approach. Trained on our TMD dataset, we ensure that the evaluation reflects diverse musical styles. As shown in Figure 11 and demonstrated in the accompanying videos, our method produces dances with better visual quality and achieves superior alignment with both text description, beat, and genre of the music, surpassing previous state-of-the-art techniques.

