# OpenReview forum: "Motion Generalist: Multimodal Motion Generation"
_ICLR.cc/2026/Conference — ICLR 2026 Conference Withdrawn Submission_

### Official Review · Reviewer_rkcE · 2025-10-29

**Soundness:** 2
**Presentation:** 3
**Contribution:** 3
**Rating:** 6
**Confidence:** 4

**Summary:**

This paper studies multimodal motion generation conditioned on text and music. The authors identify a limitation in masked-autoregressive methods: masks are applied randomly and do not adapt to the conditions, which prevents the model from focusing on condition-relevant motion tokens and degrades generation quality. To address this, they propose a dynamic, condition-aware masking strategy (derived from attention between condition and motion) and introduce a pseudo-labelled text+music→motion dataset for evaluation. The method shows strong results on HumanML3D and demonstrates promising control when both text and music are provided.

**Strengths:**

1. The authors propose and release a new text+music→motion dataset — a valuable resource that will likely benefit future multimodal motion-generation research.

2. The proposed condition-aware masking is an interesting direction: it directly targets the failure mode of random masking (failure to learn key tokens) and the empirical results (notably on HumanML3D) are convincing.

**Weaknesses:**

1. The overall pipeline strongly resembles recent works (e.g., MoMask, MoGenTS, SALAD): the mask mechanism + spatial–temporal attention design appears to be a combination of those prior ideas. The paper does not clearly disentangle what is genuinely novel versus what is a re-combination of existing components. Without a clearer comparison, the architectural novelty looks limited.

2. The multi-modality generalization claim (that fusing modalities improves generation) lacks sufficient ablation. Table 8 only compares w/o text vs. text+music conditions. The paper should include cross-task ablations (train on text only → test on text; train on music only → test on music; train on text+music → evaluate with each single modality and with both) to quantify how modalities help or interfere.

**Questions:**

1. In Figure 3(b), after the VQ-VAE encoder produces tokens and masks are applied, are the masked tokens fed into the Temporal Adaptive Transformer or not? The figure and text are ambiguous on this point — please clarify the exact token flow.

2. The idea of deriving masks from the attention map between condition and motion is interesting. Can the authors characterize which motion tokens are typically masked by this mechanism (e.g., short transient frames, repeated/redundant frames, onset/offset frames)? A qualitative example or a visualization of masked vs. unmasked frames on real clips would help understand what the mask is selecting.

3. The first claimed contribution mentions “autoregressive masked methods,” yet the paper appears to adopt only a masked prediction training objective and does not switch to an autoregressive generation paradigm. Why did the authors not adopt an autoregressive generation strategy? What are the practical or theoretical reasons for that choice?

4. The VQ-VAE component is not described in sufficient detail. Please provide architecture specifics (encoder/decoder structure, codebook size, commitment loss weighting, whether codebooks are shared per joint/frame or global, tokenization stride, etc.).

5. For different conditioning modalities, the paper specifies how conditioning is injected in the spatial and temporal attention blocks (e.g., using self-attention or cross-attention). However, I wonder whether the authors have conducted any ablation studies to evaluate the impact of these different conditioning injection strategies on model performance.

---

### Official Review · Reviewer_S3cR · 2025-10-29

**Soundness:** 3
**Presentation:** 3
**Contribution:** 2
**Rating:** 2
**Confidence:** 5

**Summary:**

This paper proposes a multi-modal motion generation framework that combines text-to-motion and music-to-motion generation into a unified model. Additionally, the paper introduces a TMD dataset for multi-modal motion generation. The authors present quantitative experiments and some visualizations to support their claims.

**Strengths:**

1. The integration of text-to-motion and music-to-motion into a single framework is technically sound and may provide a foundation for future multi-modal motion generation research.

2. The paper is well-written and the overall method is presented clearly.

3. The idea of guiding the conditional modality to select key frames (as shown in Figure 4) is intuitive and could be valuable if demonstrated more thoroughly.

**Weaknesses:**

1. From the perspective of task novelty or network architecture, the contributions are kind of incremental. The main novelty is technically combining text and music modalities, which has been explored a lot. Therefore, the broader impact on the community is unclear.

2. Figure 4’s visualization of key-frame selection is not fully convincing on its own. Without corresponding demos or more detailed explanation (e.g., map dimensions, color bar interpretation), it is hard for the reader to judge whether the model truly identifies key frames.

3. The TMD dataset is claimed as a third contribution, but its description in the main text and appendix is very limited. The supplementary material does not include any data examples. Key aspects such as music clip length, diversity, coverage, and the use of Stable Audio Open for generating music are not thoroughly detailed. The paper only mentions human expert assessment without providing quantitative or visual evidence. Experiments on TMD are also quite limited, and some ablation studies could be conducted on TMD to better validate its quality. Without this, the claimed contribution of TMD is questionable.

4. Supplementary videos show artifacts such as mesh penetration and foot sliding. There is no direct visual comparison with baseline methods; improvements reported via metrics alone may not reflect real qualitative gains. Besides, given the large number of existing motion generation methods, the paper provides very limited comparisons and visualizations. From my own experience running some baselines, reported improvements may not translate to significant perceptual differences. Therefore, it is difficult to evaluate the actual contribution of this paper based on the current submission.

5. No failure cases or limitations are discussed.

**Questions:**

The implementation details are insufficient. It’s unclear how the model was trained, on which datasets, how many models were trained, and whether there is a separate model for each modality.

Without detailed descriptions, it is hard to evaluate the paper's contribution and reproduce it.

---

### Official Review · Reviewer_qXj9 · 2025-10-31

**Soundness:** 3
**Presentation:** 2
**Contribution:** 2
**Rating:** 4
**Confidence:** 3

**Summary:**

This paper presents Motion Generalist, a unified framework for multimodal human motion generation that supports text-, music-, and text-plus-music-conditioned synthesis. The method introduces an Attention-based Mask Modeling mechanism within an autoregressive framework to emphasize key frames and joints guided by multimodal cues. Two core modules—the Temporal Adaptive Transformer (TAT) and Spatial Aligning Transformer (SAT)—align motion both temporally and spatially with the conditioning signals. The authors also contribute a new Text-Music-Dance (TMD) dataset with 2,153 triplets, twice the size of AIST++. Experiments on HumanML3D, KIT-ML, AIST++, and TMD demonstrate consistent performance gains, achieving up to a 15% FID improvement over prior methods.

**Strengths:**

1. Novel formulation: The attention-guided masking strategy effectively overcomes the limitations of random masking and enhances controllability.

2. Comprehensive evaluation: Extensive quantitative and ablation studies strongly support the method’s effectiveness.

3. Valuable dataset: The proposed TMD dataset fills a gap in multimodal motion research and provides a solid benchmark for future work.

4. State-of-the-art performance: The method achieves leading results on HumanML3D and KIT-ML datasets.

**Weaknesses:**

1. The architectural details of the Temporal Adaptive Transformer and Spatial Aligning Transformer (Figure 3) are not clearly explained, making it difficult to grasp their design and interaction.

2. The authors do not evaluate whether the proposed TMD dataset complements existing datasets. Does training on TMD improve generalization to HumanML3D or KIT-ML?

3. Part of the TMD music is synthesized rather than real, which could introduce bias or limit generalization to natural musical inputs.

**Questions:**

1. Are there qualitative failure cases, especially for text-music synchronization?

---

### Official Review · Reviewer_NtEu · 2025-10-31

**Soundness:** 2
**Presentation:** 3
**Contribution:** 2
**Rating:** 4
**Confidence:** 5

**Summary:**

The paper proposes Motion Generalist, a unified framework for multimodal motion generation that handles text-, music-, and text&music-conditioned motion within one model. It introduces an attention-based mask modeling strategy to focus on key frames and body parts guided by the conditioning modality, implemented via two Temporal Adaptive Transformer and Spatial Aligning Transformer. A new Text-Music-Dance (TMD) dataset is also established. Extensive quantitative experiments on HumanML3D, AIST++, and TMD show consistent performance gains and better controllability than prior works.

**Strengths:**

1. The new text-music-dance dataset enrich good quality paired data sources for multimodal motion generation, which support further exploration along this direction.

1. The proposed framework effectively integrates text-to-motion, music-to-dance, and text-music-to-dance generation into a single model, demonstrated with quantitative and qualitative results. The qualitative demos of the generation results are impressive, which is a highlight of this paper.

**Weaknesses:**

1. Some key implementation details are missing in the paper, e.g. the motion representation, the design of VQ-VAE and motion tokenization.  For example, what exactly the “spatial dimension” stands for when operating the motion tokens here? The Spatial Aligning Transformer is claimed to align spatial pose with spatial condition by masking and restoring the tokens along the “spatial dimension”. However, it’s actually confusing how the motion features of different body parts can be explicitly / implicitly represented along this “spatial dimension” without any related definition.

1. The training pipeline is not clearly illustrated. What kind of objective functions is the proposed model trained with? Are the tokens masked at different positions in each different spatial / temporal layer? If so, what would be account for the final predictions for calculating the objective function? Similarly, the inference pipeline is not clear neither.

1. The qualitative figures (e.g. Figure 5, Figure 6) stack too many overlapping SMPL frames, which makes it hard for the readers to evaluate the generated motion details. Consider bigger intervals with fewer frames when sampling the motion sequences. Also, the axe or reference for temporal direction should be mentioned alongside.

1. There are still several aspects of the design choice untouched in the ablation studies. For example, can the masking ratio of the attention-based masking be a scheduled varying ratio instead of a fixed one? What if the order of stacking the spatial transformer and temporal transformer is inverted?

1. Currently, the generated dances of the two tasks seem to be biased to hiphop, krump and breakdance style. The text prompts given are also biased to describe hiphop and street dance movements. Besides, the generated musics are very similar, both of 120-130 bpm, similar beats and styles. It would be more convincing to show broader range of musics and text prompts.

**Questions:**

Why is the length for text-to-motion and music-to-motion fixed to 6 sec per clip, while when applied to 4D avatars it is extended to 10 sec? Is there any particular design choice or algorithm to deal with variable length generation?

Please refer to the weaknesses section for more details. The actual score I would like to give for now is around 5. So if the identified issues and questions are properly addressed, I would consider revising my score to a more positive level.

---

### Note · Authors · 2025-11-12

I have read and agree with the venue's withdrawal policy on behalf of myself and my co-authors.